# Hybrid-Collaborative Augmentation and Contrastive Sample Adaptive-Differential Awareness for Robust Attributed Graph Clustering

**Tianxiang Zhao**[1] **Youqing Wang**[1] **Jinlu Wang**[2] **Jiapu Wang**[2]
**Mingliang Cui**[1] **Junbin Gao**[3] **Jipeng Guo**[1]*

[1]College of Information Science and Technology, Beijing University of Chemical Technology
[2]Beijing University of Technology
[3]The University of Sydney Business School, The University of Sydney
[1]{tianxiangzhao, 2021400208, guojipeng}@buct.edu.cn, wang_youqing@mail.buct.edu.cn
[2]wangjinlu@emails.bjut.edu.cn, jiapuwang9@gmail.com
[3]junbin.gao@sydney.edu.au

## Abstract

Due to its powerful capability of self-supervised representation learning and clustering, contrastive attributed graph clustering (CAGC) has achieved great success, which mainly depends on effective data augmentation and contrastive objective setting. However, most CAGC methods utilize edges as auxiliary information to obtain node-level embedding representation and only focus on node-level embedding augmentation. This approach overlooks edge-level embedding augmentation and the interactions between node-level and edge-level embedding augmentations across various granularity. Moreover, they often treat all contrastive sample pairs equally, neglecting the significant differences between hard and easy positive-negative sample pairs, which ultimately limits their discriminative capability. To tackle these issues, a novel robust attributed graph clustering (RAGC), incorporating hybrid-collaborative augmentation (HCA) and contrastive sample adaptive-differential awareness (CSADA), is proposed. First, node-level and edge-level embedding representations and augmentations are simultaneously executed to establish a more comprehensive similarity measurement criterion for subsequent contrastive learning. In turn, the discriminative similarity further consciously guides edge augmentation. Second, by leveraging pseudo-label information with high confidence, a CSADA strategy is elaborately designed, which adaptively identifies all contrastive sample pairs and differentially treats them by an innovative weight modulation function. The HCA and CSADA modules mutually reinforce each other in a beneficent cycle, thereby enhancing discriminability in representation learning. Comprehensive graph clustering evaluations over six benchmark datasets demonstrate the effectiveness of the proposed RAGC against several state-of-the-art CAGC methods. The code of RAGC could be available at https://github.com/TianxiangZhao0474/RAGC.git.

## 1 Introduction

Attributed graph representation learning, as an effective method, fully leverages both attribute features and the neighborhood structure among samples [1]. Graph data is widely present in practice, making graph representation learning essential across various fields such as node classification [2, 3, 4, 5],

---

*Corresponding author: `Jipeng Guo`

39th Conference on Neural Information Processing Systems (NeurIPS 2025).

node clustering [6, 7, 8, 9], knowledge graph analysis [10], traffic prediction [11], due to its powerful representation capability. Attributed Graph Clustering (AGC), which uses Graph Neural Networks (GNNs) to learn discriminative embedding representation [12, 13] and partitions nodes into disjoint clusters without label information, has become a popular and fundamental task in the field of data mining. However, it also faces numerous challenges [14].

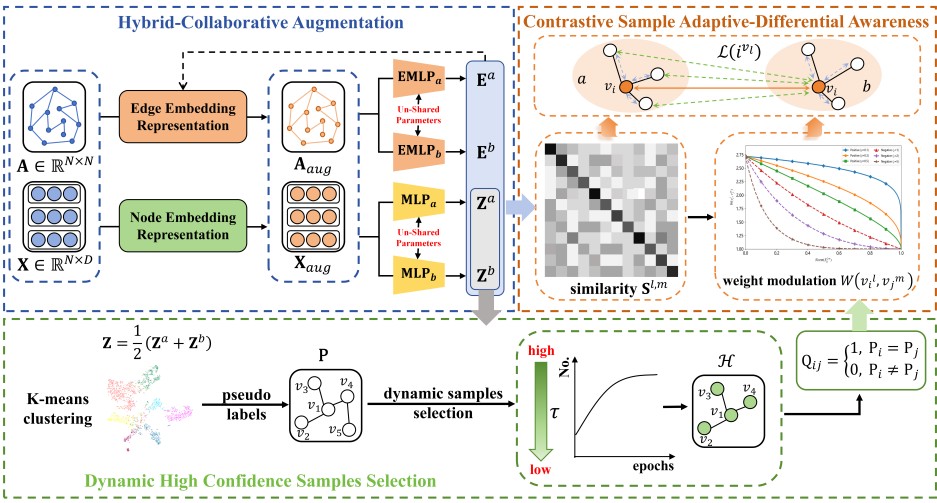

Figure 1: The overall framework of the proposed RAGC, which mainly consists of HCA and CSADA modules. The HCA module constructs reliable augmented views and comprehensive similarity by hybrid-collaborative augmentation from node-level embedding and edge-level embedding. With dynamic high confidence samples selection strategy, the CSADA module significantly distinguishes the contrastive samples through a powerful weight modulation function.

Recently, contrastive learning, a powerful self-supervised representation learning paradigm that does not rely on label information, has attracted widespread attention and achieved significant progress in various fields, such as computer vision [15], natural language processing [16], feature selection [17]. The key idea of contrastive learning is to generate supervised signals from the data itself by data augmentation and then learn discriminative representation by optimizing a contrastive objective between positive and negative sample pairs. Building upon the prominent success of contrastive learning, several contrastive AGC (CAGC) methods have integrated contrastive learning into the AGC framework to boost clustering performance. Specifically, most existing CAGC methods aim to construct augmented views using carefully designed augmentation operators, encouraging identical samples across different views to align while ensuring distinct samples diverge [18].

Although CAGC methods have achieved satisfactory performance, data augmentation and contrastive objective design remain critical components that significantly impact on clustering performance. In graph data augmentation, various manual strategies introduce perturbations to node attributes or structure, such as attribute masking [19], edge perturbation [20], noise perturbation [21]. Furthermore, to improve the adaptability and flexibility of data augmentation for downstream tasks, several learnable strategies have been proposed [22, 23]. However, most existing CAGC methods only treat edges as auxiliary information to obtain augmented graph embedding representations and measure contrastive similarity between samples by single embedding representation-level augmentations, while paying no attention to the importance of *edge-level embedding representation learning and augmentation*. Few existing CAGC methods have explored the *collaborative interaction* between node-level and edge-level embedding augmentations across different granularity. Moreover, the contrastive objective setting also plays an important role in CAGC methods. Recently, contrastive sample awareness strategies have attracted great attention, enabling more exhaustive contrastive learning and improving discriminative capability. However, most existing CAGC methods *only perceive a subset of contrastive sample pairs in a differentiated manner*, often prioritizing specific cases, such as hard negative samples [24] or hard samples [25, 26, 27]. Specifically, they struggle to *adaptively and effectively distinguish all contrastive samples, especially in differentiating positive-*

*easy, positive-hard, negative-easy, and negative-hard ones*. This limitation compromises the quality of embedding representation learning.

To remedy these shortcomings, this study proposes a novel **R**obust **A**ttributed **G**raph **C**lustering (**RAGC**), which goes beyond single node-level embedding augmentation and partial contrastive sample awareness, offering a more comprehensive approach to contrastive graph clustering. The RAGC seeks to leverage hybrid augmentations of node-level embedding representation and edge-level representation through collaborative interaction, while also incorporating comprehensive contrastive sample awareness with adaptive and differential weighting.

As shown in Fig. 1, the overall framework of RAGC mainly consists of two key modules: Hybrid-Collaborative Augmentation (HCA) and Contrastive Sample Adaptive-Differential Awareness (CSADA) modules. In particular, the HCA module simultaneously constructs node-level embedding representation augmentation and edge-level embedding representation augmentation to provide a more comprehensive and reliable contrastiveness metric. And, the cross-view contrastive similarity matrix is leveraged to refine and optimize the graph structure to achieve the multi-level contrastiveness and collaborative augmentation between node-level embedding representation and edge-level embedding representation and improve robustness against graph structure noise. Furthermore, guided by clustering pseudo labels with high confidence and adaptive confidence factor, the CSADA module thoroughly identifies and perceives the contrastive samples to enhance the discriminative capability of representation learning. Specifically, contrastive samples with high confidence are treated differently based on positive-negative and hard-easy perspectives through a weight modulation function, which adaptively adjusts participation of samples in self-supervised training. By leveraging contrastive similarity and clustering pseudo labels as a bridge, the HCA and CSADA modules mutually promote each other, further enhancing representation learning.

For clarity, the main contributions of this study are summarized as follows:

- Instead of solely relying on node-level embedding, this study further explores edge-level embedding representation augmentation and unifies them to obtain more comprehensive data augmentation and a reliable cross-view contrastive similarity across multiple granularity-levels.

- Guided by relatively reliable clustering pseudo-label information with dynamic confidence, contrastive sample pairs are adaptively differentiated by a novel and well-designed weight modulation function. To the best of the authors' knowledge, this study is the first to simultaneously perceive and emphasize positive-easy, positive-hard, negative-easy, and negative-hard samples, all of which contribute to deeper contrastiveness and improve the discriminative ability of self-supervised learning.

- Extensive experimental results on six benchmark datasets clearly demonstrate the superiority of the proposed RAGC over representative state-of-the-art methods. More importantly, the strong scalability of CSADA strategy in enhancing other CAGC methods is also validated.

## 2 Model Formulation

### 2.1 Notations

For an attributed graph $\mathcal{G} = \{\mathcal{V}, \mathcal{E}, \mathbf{X}\}$, $\mathcal{V} = \{v_1, \cdots, v_N\}$ represents the node set with $N$ nodes from disjoint $K$ classes, $\mathcal{E}$ is the edge set describing the connection relationships between pairwise nodes, and $\mathbf{X} \in \mathbb{R}^{N \times D}$ denotes the node attribute features. The connection relationships in the edge set $\mathcal{E}$ can be mathematically formulated as an adjacency matrix $\mathbf{A} \in \{0, 1\}^{N \times N}$, where each element $\mathbf{A}_{ij}$ is defined as follows:

$$\mathbf{A}_{ij} = \begin{cases} 1, & \text{if } (v_i, v_j) \in \mathcal{E} \\ 0, & \text{otherwise} \end{cases} \tag{1}$$

The diagonal degree matrix is $\mathbf{D} = \mathbf{diag}(d_1, \cdots, d_N)$, where $d_i = \sum_{j=1}^{N} \mathbf{A}_{ij}$. The normalized graph Laplacian matrix is defined as $\widetilde{\mathbf{L}} = \mathbf{I} - \widetilde{\mathbf{A}}$, where $\widetilde{\mathbf{A}} = \widehat{\mathbf{D}}^{-1/2} \widehat{\mathbf{A}} \widehat{\mathbf{D}}^{-1/2}$ is the normalized adjacency

Table 1: The explanation of main notations.

| Notation | Meaning |
|---|---|
| $\mathbf{X} \in \mathbb{R}^{N \times D}$ | Original attribute feature matrix |
| $\mathbf{A} \in \mathbb{R}^{N \times N}$ | Original adjacency matrix |
| $\widehat{\mathbf{A}} \in \mathbb{R}^{N \times N}$ | Modified adjacency matrix with self-connections |
| $\mathbf{X}_{aug} \in \mathbb{R}^{N \times D}$ | Node-level embedding representation |
| $\mathbf{A}_{aug} \in \mathbb{R}^{N \times N}$ | Dynamic semantic correlation matrix |
| $\mathbf{I} \in \{0, 1\}^{N \times N}$ | Identity matrix |
| $\mathbf{D}, \widehat{\mathbf{D}} \in \mathbb{R}^{N \times N}$ | Degree matrices |
| $\widetilde{\mathbf{L}} \in \mathbb{R}^{N \times N}$ | Normalized graph Laplacian matrix |
| $\mathbf{Z}^a, \mathbf{Z}^b \in \mathbb{R}^{N \times d}$ | Node-level embeddings in augmented views |
| $\mathbf{E}^a, \mathbf{E}^b \in \mathbb{R}^{N \times d}$ | Edge-level embeddings in augmented views |
| $\mathbf{Z} \in \mathbb{R}^{N \times d}$ | Fused node-level embedding representation |
| $\mathbf{P} \in \{1, \cdots, K\}^N$ | Clustering pseudo labels |
| $\mathbf{Q} \in \{0, 1\}^{N \times N}$ | Pseudo label semantic correlation matrix |
| $\mathcal{H} \subseteq \mathcal{V}, |\mathcal{H}| = M$ | High-confidence sample set with $M$ nodes |

matrix with the refined adjacency matrix of self-connections $\widehat{\mathbf{A}} = \mathbf{A} + \mathbf{I}$ and $\widehat{\mathbf{D}} = \mathbf{D} + \mathbf{I}$. For clarity, the notations used in this study are summarized and explained in Table 1.

## 2.2 Hybrid-Collaborative Augmentation

Learning edge representation is beneficial for effectively improving the discriminability of embedding representation. Hence, node-level embedding representation and edge representation encoders embed the attributed graph into the various latent spaces, while two-level augmentations are simultaneously employed to construct a comprehensive contrastive similarity metric.

**Node-level Embedding Augmenter.** To achieve simple and effective node-level embedding augmentation, a representation-first, augmentation-later strategy is adopted. To improve representation generalization ability and robustness to noise, a mixed attribute perturbation augmentation strategy is employed, where Gaussian noise and random masking are added to the original attribute features, i.e.,

$$\mathbf{X}_N = \mathbf{X} + \mathcal{N}(0, \sigma_N) \tag{2}$$
$$\mathbf{X}_M = \mathbf{X} \odot \mathcal{M}_{mask}(r) \tag{3}$$

where $\mathcal{N}(0, \sigma_N)$ and $\mathcal{M}_{mask}(r)$ indicate the Gaussian noise with standard deviation $\sigma_N$ and the randomly generated mask matrix with mask ratio $r$, respectively. After that, a multi-order low-passing graph Laplacian filter is performed on both the noisy attribute matrices $\mathbf{X}_N$ and $\mathbf{X}_M$, effectively suppressing high-frequency noise and aggregating neighbor information:

$$\mathbf{X}_N^L = (\mathbf{I} - \widetilde{\mathbf{L}})^{t_n} \mathbf{X}_N, \mathbf{X}_M^L = (\mathbf{I} - \widetilde{\mathbf{L}})^{t_m} \mathbf{X}_M \tag{4}$$

where $\mathbf{I} - \widetilde{\mathbf{L}}$ is the graph Laplacian filter, and $t_n$ and $t_m$ are the order of filters. Furthermore, $\mathbf{X}_N^L$ and $\mathbf{X}_M^L$ are fused to obtain the robust and comprehensive node-level embedding representation $\mathbf{X}_{aug}$ for subsequent contrastive augmentation:

$$\mathbf{X}_{aug} = \frac{1}{2} \left( \mathbf{X}_N^L + \mathbf{X}_M^L \right) \tag{5}$$

Then, two MLPs encoders are employed on the node-level embedding representation $\mathbf{X}_{aug}$ to generate two augmented views:

$$\mathbf{Z}^a = \mathbf{MLP}_a \left( \mathbf{X}_{aug} \right), \mathbf{Z}^b = \mathbf{MLP}_b \left( \mathbf{X}_{aug} \right) \tag{6}$$

where $\mathbf{MLP}_a$ and $\mathbf{MLP}_b$ are two simple and learnable augmenters with the same MLP structure but without shared parameters. After that, augmented node-level embedding representations $\mathbf{Z}^a$ and $\mathbf{Z}^b$ are further normalized along the row dimension by $l_2$-norm to facilitate the calculation of contrastive similarity:

$$\mathbf{Z}^a (i, :) = \mathbf{Z}^a (i, :) / \|\mathbf{Z}^a (i, :)\|_2, \mathbf{Z}^b (i, :) = \mathbf{Z}^b (i, :) / \|\mathbf{Z}^b (i, :)\|_2 \tag{7}$$

**Edge-level Embedding Augmenter.** The multi-order graph filter is prone to over-smoothing. To effectively leverage the structural information of the attributed graph, the edge-level embedding representation and augmentations can be obtained using two MLPs encoders:

$$\mathbf{E}^a = \mathbf{EMLP}_a\left(\mathbf{A}\right), \mathbf{E}^b = \mathbf{EMLP}_b\left(\mathbf{A}\right) \tag{8}$$

where, similarly, $\mathbf{EMLP}_a$ and $\mathbf{EMLP}_b$ are two structure encoders with the same MLP structure but without shared parameters. The $\mathbf{E}^a$ and $\mathbf{E}^b$ are augmented edge-level embedding representations, which contains different and rich semantic information. Further, the $\mathbf{E}^a$ and $\mathbf{E}^b$ are also normalized along the row dimension:

$$\mathbf{E}^a\left(i,:\right) = \mathbf{E}^a\left(i,:\right)/\|\mathbf{E}^a\left(i,:\right)\|_2, \mathbf{E}^b\left(i,:\right) = \mathbf{E}^b\left(i,:\right)/\|\mathbf{E}^b\left(i,:\right)\|_2 \tag{9}$$

**Comprehensive Contrastive Similarity Construction and Collaborative Interaction between Various Augmentations.** With node-level embedding augmentations $\mathbf{Z}^a$, $\mathbf{Z}^b$ and edge-level embedding augmentations $\mathbf{E}^a$, $\mathbf{E}^b$, the comprehensive contrastive similarity matrix across attribute and structure is constructed as follows:

$$\mathbf{S}^{l,m} = \alpha \mathbf{Z}^l\left(\mathbf{Z}^m\right)^T + (1-\alpha)\mathbf{E}^l\left(\mathbf{E}^m\right)^T, \forall\, l,m \in \{a,b\} \tag{10}$$

where $\alpha$ is a learnable balance coefficient that adjusts the role between the node-level and edge-level embeddings.

Obviously, the comprehensive similarity matrices $\mathbf{S}^{l,m}$ will provide beneficial guidance for contrastive training. In addition, the structural information in similarity matrix $\mathbf{S}^{a,b}$ is more discriminative and can provide adaptive refinement for edge-level embedding representation learning. Eq. (8) is further formulated as follows:

$$\mathbf{E}^a = \mathbf{EMLP}_a\left(\mathbf{A}_{aug}\right), \mathbf{E}^b = \mathbf{EMLP}_b\left(\mathbf{A}_{aug}\right) \tag{11}$$

where

$$\mathbf{A}_{aug} := Norm\left(\mathbf{Z}^a(\mathbf{Z}^b)^T + \mathbf{E}^a(\mathbf{E}^b)^T\right) \odot \mathbf{A}_{aug} \tag{12}$$

where $Norm(\cdot)$ is the min-max normalization operator. The $\mathbf{A}_{aug}$ is a dynamic semantic correlation matrix and iteratively updated to achieve structure rectification, and the initial setting is $\mathbf{A}_{aug} = \mathbf{A}$. The node-level embedding representation augmentations $\mathbf{Z}^a$ and $\mathbf{Z}^b$ also guide the edge-level embedding augmentations in reverse, achieving the collaborative interaction between node-level embedding augmentations and edge-level embedding augmentations.

### 2.3 Contrastive Sample Adaptive-Differential Awareness

Contrastive learning usually maximizes the agreement between different augmentations of the same objective sample while minimizing the similarity between the augmented views from different objective samples. For target sample $v_i$ in $l$-th augmented view, the InfoNCE loss is utilized for self-supervised representation learning, i.e.,

$$\mathcal{L}(v_i^l) = -\log \frac{\sum\limits_{m \neq l} e^{\theta(v_i^l, v_i^m)}}{\sum\limits_{m \neq l} e^{\theta(v_i^l, v_i^m)} + \sum\limits_{j \neq i}\sum\limits_{m \in \{a,b\}} e^{\theta(v_i^l, v_j^m)}} \tag{13}$$

where $\theta(v_i^l, v_j^m)$ is the contrastive similarity, defined as a similar way in Eq. (10). However, $\mathbf{S}^{l,m}$ contains edge-level embedding augmentations, so it is a more comprehensive similarity. Further, it is evident that the classical InfoNCE loss treats all contrastive sample pairs equally limiting the discriminability of self-supervised representation learning. To deepen contrastiveness, contrastive samples with high-confidence clustering structure are given greater emphasis and effectively distinguished by a dynamic weighting strategy.

**Dynamic High Confidence Samples Selection.** In contrastive learning, focusing on high confidence samples with clear clustering structure helps improve the boundaries of representation learning. After node-level embedding representation augmentation, a comprehensive, clustering-oriented representation of node-level embedding is obtained by linearly fusing $\mathbf{Z}^a$ and $\mathbf{Z}^b$:

$$\mathbf{Z} = \frac{1}{2}(\mathbf{Z}^a + \mathbf{Z}^b) \tag{14}$$

Then, the K-means clustering algorithm is performed on $\mathbf{Z}$ to extract clustering information, including pseudo labels $\mathbf{P} \in \{1, \cdots, K\}^N$ and clustering centers $\{\mathbf{K}_1, \cdots, \mathbf{K}_K\}$. To select high confidence samples more efficiently, the confidence score $\mathbf{CONF}_i$ is defined based on the distance between the embedding and the corresponding center:

$$\mathbf{CONF}_i = \sigma\left(\mathcal{D}(\mathbf{Z}_i, \mathbf{K}_{P_i})\right) \tag{15}$$

where $\mathbf{K}_{P_i}$ is the clustering center corresponding to sample $\mathbf{Z}_i$, $\mathcal{D}(\mathbf{Z}_i, \mathbf{K}_{P_i})$ is the distance function, and $\sigma$ is the $softmax$ activation function to normalize the distance. The high confidence sample set is constructed by selecting top $M$ samples with the highest confidence scores:

$$\mathcal{H} = \{v_i|\, top\,(\mathbf{CONF}, M)\} \tag{16}$$

where $M = [N(1 - \tau)]$ is the number of high confidence samples, $[\cdot]$ is rounding function, and $\tau$ is a dynamic confidence factor. In practice, the confidence parameter $\tau$ is initially set to a large value, and then gradually decreases as the training progresses. This ensures that an increasing number of samples are selected and focused on, which is beneficial for contrastive training. With iterative training, the discriminability of the embedding representation $\mathbf{Z}$ and clustering structure improve. Therefore, more high confidence samples are prioritized in contrastive learning, further enhancing self-supervised representation learning in a reinforcing cycle.

In addition, the label semantic correlation matrix $\mathbf{Q} \in \{0, 1\}^{N \times N}$ is constructed based on clustering pseudo labels $\mathbf{P}$ as follows:

$$Q_{ij} = \begin{cases} 1, & P_i = P_j \\ 0, & P_i \neq P_j \end{cases} \tag{17}$$

$Q_{ij}$ effectively reveals the pseudo label correlation between nodes $v_i$ and $v_j$. Specifically, when $Q_{ij} = 1$, it indicates that nodes $v_i$ and $v_j$ belong to the same cluster with high probability, i.e., they are more likely to be positive sample pairs. Conversely, when $Q_{ij} = 0$, it indicates that the nodes $v_i$ and $v_j$ have different pseudo labels, implying that they are more likely to be negative sample pairs.

**Adaptive-Differential Weight Modulation Function for Contrastive Sample.** Different contrastive sample pairs play significantly different roles in self-supervised training. For example, some samples are easily confused and require deliberate attention, which helps enhance the discriminative ability of representation learning. To adaptively distinguish contrastive sample pairs, a novel weight modulation function is designed and defined as:

$$W(v_i^l, v_j^m) = \begin{cases} 1, & v_i, v_j \notin \mathcal{H} \\ e^{\left(1 - Norm(\mathbf{S}_{ij}^{l,m})\right)^\beta}, & v_i, v_j \in \mathcal{H}, Q_{ij} = 1 \\ e^{\left(1 - Norm(\mathbf{S}_{ij}^{l,m})\right)^\gamma}, & v_i, v_j \in \mathcal{H}, Q_{ij} = 0 \end{cases} \tag{18}$$

where $\beta \in (0, 1)$ and $\gamma \in [1, 5]$ are adjustable weight factors. Fig. (2) visualizes the weight modulation function under various cases. Some desirable properties of $W(v_i^l, v_j^m)$ can be summarized as follows, all of which contribute to enhancing contrastive learning.

1) For high-confidence contrastive samples, $W(v_i^l, v_j^m) \geq 1$ is always satisfied, ensuring strong attention to contrastive samples with a clear clustering structure. In contrast, samples with an unclear clustering structure should not be emphasized, as it may lead to an inaccurate label semantic correlation matrix $\mathbf{Q}$, which could negatively impact contrastive learning.

2) For a positive sample pair (i.e., $Q_{ij} = 1$), the greater the similarity, the easier it is to promote agreement between them. Hence, positive hard sample pairs with low similarity are up-weighted, while positive easy sample pairs with high similarity still receive a relatively high weight. And, the weight assigned to positive hard sample pairs is always greater than the weight of positive easy ones. For a negative sample pair (i.e., $Q_{ij} = 0$), the greater the similarity, the more difficult it is to push them apart. Hence, negative hard sample pairs with high similarity are down-weighted, ensuring that they are forced to separate as much as possible.

3) For contrastive similarity, positive sample pairs are assigned higher weights to strengthen the aggregation of similar samples. In addition, under the same similarity, negative sample pairs are assigned relatively lower weights than positive ones, thereby pushing apart negative samples more effectively during optimization. Hence, appropriately adjusting $\beta \in (0, 1)$ and $\gamma \in [1, 5]$ allows

control over the weight difference between positive and negative sample pairs, enhancing the flexibility of the weight modulation mechanism in dealing with various applications. Specifically, reducing $\beta$ and increasing $\gamma$ sufficiently amplifies the weight difference between positive and negative sample pairs with moderate similarity, thereby strengthening the ability of RAGC to capture complex boundaries and improve discriminative representation learning.

In summary, the CSADA module adaptively distinguishes contrastive sample pairs by the well-designed weight modulation function $W(v_i^l, v_j^m)$, effectively handling positive-hard, positive-easy, negative-hard, and negative-easy sample pairs.

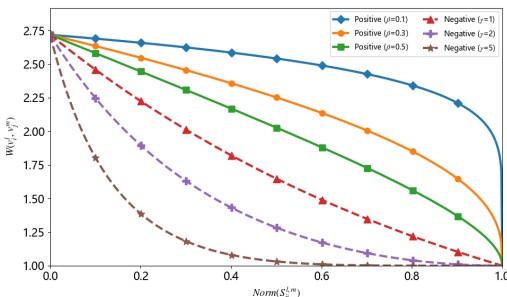

Figure 2: Visualization of weight modulation function $W(v_i^l, v_j^m)$ with different weight adjustment factors.

## 2.4 Objective Function

By calculating the view-wise contrastive similarity function $\mathbf{S}^{l,m}$ and the weight adjustment function $W(v_i^l, v_j^m)$, for a target sample $v_i$ in $l$-th view, the contrastive sample adaptive-differential awareness guided self-supervised training loss can be defined as:

$$\mathcal{L}(v_i^l) = -log \frac{\sum_{m \neq l} e^{W(v_i^l, v_i^m)\mathbf{S}_{ii}^{l,m}}}{\sum_{m \neq l} e^{W(v_i^l, v_i^m)\mathbf{S}_{ii}^{l,m}} + \sum_{j \neq i} \sum_{m \in \{a,b\}} e^{W(v_i^l, v_j^m)\mathbf{S}_{ij}^{l,m}}} \tag{19}$$

By incorporating a comprehensive contrastive similarity metric $\mathbf{S}^{l,m}$ and an effective weight adjustment mechanism, the RAGC enforces discriminative learning across positive-hard, positive-easy, negative-hard, and negative-easy sample pairs. The overall training loss of the RAGC method is formulated as follows:

$$\mathcal{L} = \frac{1}{2N} \sum_{l \in \{a,b\}} \sum_{i=1}^{N} \mathcal{L}(v_i^l) \tag{20}$$

For clarity, the detailed training process of RAGC is described in Algorithm 1.

## 3 Experiment

In this section, the superiority and effectiveness of the proposed RAGC are validated through extensive experiments. Detailed experimental settings are shown in Appendix B, and additional results are shown in Appendix C.

## 3.1 Experimental Setting

Following the key baseline in [26], the proposed RAGC is evaluated on six challenging graph benchmark datasets, including CORA [28], CITESEER (CITE) [28], AMAP [29], BAT[21], EAT [21] and UAT [21]. The detailed statistics of these datasets are briefly summarized in Appendix B.1. In the experiment, RAGC is compared against 11 state-of-the-art baseline methods, which are introduced in the Appendix B.2. Moreover, to reduce the impact of random initialization, the parameters are

---

**Algorithm 1** Training process of RAGC.

---

**Input:** Attributed graph $\mathcal{G}$; Cluster number $K$; Iteration number $It$; Standard deviation of Gaussian noise $\sigma_N$; Mask ratio $r$; Filter orders $t_n$, $t_m$; Hyper-parameters $\tau$, $\beta$, $\gamma$.

**Output:** Clustering label matrix $\mathbf{P}$.

1: Construct the node-level embedding representation $\mathbf{X}_{aug}$ using Eqs. (2)-(5).
2: Initialize $\mathbf{A}_{aug} = \mathbf{A}$.
3: **for** $iter = 1$ to $It$ **do**
4:  Obtain two augmented views of node-level embedding representation $\mathbf{X}_{aug}$ using Eqs. (6)-(7).
5:  Obtain augmented views of edge-level embedding representation using Eqs. (9), (11).
6:  Update the augmented structure matrix $\mathbf{A}_{aug}$ using Eq. (12).
7:  Calculate contrastive similarity $\mathbf{S}^{l,m}$ using Eq. (10).
8:  Perform K-means on comprehensive node-level embedding $\mathbf{Z}$ to obtain clustering pseudo labels $\mathbf{P}$ and construct dynamic high-confidence sample set $\mathcal{H}$ using Eqs. (14)-(16).
9:  Obtain label semantic correlation matrix $\mathbf{Q}$ based on pseudo labels $\mathbf{P}$ using Eq. (17).
10:  Calculate the weight modulation function $W(v_i^l, v_j^m)$ to distinguish contrastive sample pairs using Eq. (18).
11:  Train the entire network by minimizing $\mathcal{L}$ in Eq. (20).
12: **end for**
13: Calculate the final clustering result $\mathbf{P}$ by performing K-means on $\mathbf{Z}$.

---

initialized with different random seeds and all methods run 10 times, and then the experimental results report the mean value and corresponding standard deviation for all clustering metrics.

## 3.2 Performance Comparison

The quantitative experimental results of the all AGC comparative methods on the benchmark datasets are shown in Table 2. Notably, except for a few cases, the proposed RAGC achieves optimal clustering performance across all six datasets, clearly demonstrating its superiority and effectiveness. Specifically, the performance improvements of RAGC are significant on several datasets. For example, on the BAT dataset, RAGC outperforms the sub-optimal HSAN method by 2.65%, 2.53%, 2.44% and 2.53% in terms of ACC, NMI, ARI and F1 metrics, respectively. Similarly, for the CORA dataset, the RAGC exceeds the sub-optimal HSAN by 1.67%, 1.41%, 2.32% and 1.80% on ACC, NMI, ARI and F1 metrics, respectively.

Obviously, the proposed RAGC significantly outperforms classical generative-based, adversarial-based, and existing contrastive AGC methods. This effectiveness stems from its hybrid-collaborative augmentation that integrates node-level and edge-level embeddings, and a contrastive sample adaptive-differential awareness strategy that distinguishes between hard-easy and positive-negative sample pairs. These modules enhance the discriminability and boundary awareness of the learned representations, leading to improved clustering results. Experimental comparisons, including with strong baselines like HSAN, confirm RAGC's effectiveness in contrastive learning for graph clustering. A more detailed analysis can be found in the Appendix C.1.

## 3.3 Ablation Study

The ablation experiments are conducted to evaluate the impact of the key modules in the RAGC method. Specifically, three ablation variants of RAGC are defined as follows: 1) **(w/o) D**: This ablation variant represents RAGC with static high confidence samples selection, where the confidence coefficient $\tau$ remains fixed in the training process. 2) **(w/o) H**: This ablation variant represents RAGC without the HCA module, where a simple low-pass filtering is utilized to obtain node-level embedding representation and the corresponding augmented views are generated by MLPs encoders. 3) **(w/o) C**: This ablation variant is RAGC without the CSADA module, where the classical infoNCE contrastive loss is utilized for self-supervised training.

From the experimental results of the ablation variants in Fig. 3, several notable observations can be drawn. 1) The ablation results clearly validate the effectiveness of each core component in RAGC. Removing any key strategy leads to performance degradation, highlighting their compatibility and necessity. 2) Notably, the HCA module contributes most significantly by enabling collaborative

Table 2: The performance comparison on six datasets. All results are reported with (mean ± std) under ten runs. The red and blue values indicate the best and the suboptimal results, respectively.

| Dataset | Metric | Generative and Adversarial AGC Methods | | | | Classic CAGC Methods | | | | Hard Sample Aware CAGC Methods | | | |
|---|---|---|---|---|---|---|---|---|---|---|---|---|---|
| | | DAEGC | SDCN | DFCN | ARGA | AGE | NCLA | CCGC | GL | GDCL | ProGCL | HSAN | RAGC |
| CORA | ACC | 70.43±0.36 | 35.60±2.83 | 36.33±0.49 | 71.04±0.25 | 73.50±1.83 | 51.09±1.25 | 73.88±1.20 | 74.91±1.78 | 70.83±0.47 | 57.13±1.23 | 77.07±1.56 | 78.74±0.72 |
| | NMI | 52.89±0.69 | 14.28±1.91 | 19.36±0.87 | 51.06±0.52 | 57.58±1.42 | 31.80±0.78 | 56.45±1.04 | 58.16±0.83 | 56.60±0.36 | 41.02±1.34 | 59.21±1.03 | 60.62±0.34 |
| | ARI | 49.63±0.43 | 07.78±3.24 | 04.67±2.10 | 47.71±0.33 | 50.60±2.14 | 36.66±1.65 | 52.51±1.89 | 53.82±2.25 | 48.05±0.72 | 30.71±2.70 | 57.52±2.70 | 59.84±0.60 |
| | F1 | 68.27±0.57 | 24.37±1.04 | 26.16±0.50 | 69.27±0.39 | 69.68±1.59 | 51.12±1.12 | 70.98±2.79 | 73.33±1.86 | 52.88±0.97 | 45.68±1.29 | 75.11±1.40 | 76.91±0.78 |
| CITE | ACC | 64.54±1.39 | 65.96±0.31 | 69.50±0.20 | 61.07±0.49 | 69.73±0.24 | 59.23±2.32 | 69.84±0.94 | 70.12±0.36 | 66.39±0.65 | 65.92±0.80 | 71.15±0.80 | 71.30±0.42 |
| | NMI | 36.41±0.86 | 38.71±0.32 | 43.90±0.20 | 34.40±0.71 | 44.93±0.53 | 36.68±0.89 | 44.33±0.79 | 43.56±0.35 | 39.52±0.38 | 39.59±0.39 | 45.06±0.74 | 45.32±0.51 |
| | ARI | 37.78±1.24 | 40.17±0.43 | 45.50±0.30 | 34.32±0.70 | 45.31±0.41 | 33.37±0.53 | 45.68±1.80 | 44.85±0.69 | 41.07±0.96 | 36.16±1.11 | 47.05±1.12 | 46.51±0.63 |
| | F1 | 62.20±1.32 | 63.62±0.24 | 64.30±0.20 | 58.23±0.31 | 64.45±0.27 | 52.67±0.64 | 62.71±2.06 | 65.01±0.39 | 61.12±0.70 | 57.89±1.98 | 63.01±1.79 | 62.49±1.13 |
| AMAP | ACC | 75.96±0.23 | 53.44±0.81 | 76.82±0.23 | 69.28±2.30 | 75.98±0.68 | 67.18±0.75 | 77.25±0.41 | 77.24±0.87 | 43.75±0.78 | 51.53±0.38 | 77.02±0.33 | 78.29±0.82 |
| | NMI | 65.25±0.45 | 44.85±0.83 | 66.23±1.21 | 58.36±2.76 | 65.38±0.61 | 63.63±1.07 | 67.44±0.48 | 67.12±0.92 | 37.32±0.28 | 39.56±0.39 | 67.21±0.33 | 67.50±0.64 |
| | ARI | 58.12±0.24 | 31.21±1.23 | 58.28±0.74 | 44.18±4.41 | 55.89±1.34 | 46.30±1.59 | 57.99±0.66 | 58.14±0.82 | 21.57±0.51 | 34.18±0.89 | 58.01±0.48 | 59.53±1.39 |
| | F1 | 69.87±0.54 | 50.66±1.49 | 71.25±0.31 | 64.30±1.95 | 71.74±0.93 | 73.04±1.08 | 72.18±0.57 | 73.02±2.34 | 38.37±0.29 | 31.97±0.44 | 72.03±0.46 | 72.67±2.14 |
| BAT | ACC | 52.67±0.00 | 53.05±4.63 | 55.73±0.06 | 67.86±0.80 | 56.68±0.76 | 47.48±0.64 | 75.04±1.78 | 75.50±0.87 | 45.42±0.54 | 55.73±0.79 | 77.15±0.72 | 79.77±1.29 |
| | NMI | 21.43±0.35 | 25.74±5.71 | 48.77±0.51 | 49.09±0.54 | 36.04±1.54 | 24.36±1.54 | 50.23±2.43 | 50.58±0.90 | 31.70±0.42 | 28.69±0.92 | 53.21±0.93 | 55.74±1.70 |
| | ARI | 18.18±0.29 | 21.04±4.97 | 37.76±0.23 | 42.02±1.21 | 26.59±1.83 | 24.14±0.98 | 46.95±3.09 | 47.45±1.53 | 19.33±0.57 | 21.84±1.34 | 52.20±1.11 | 54.64±2.24 |
| | F1 | 52.23±0.03 | 46.45±5.90 | 50.90±0.12 | 67.02±1.15 | 55.07±0.80 | 42.25±0.34 | 74.90±1.80 | 75.40±0.88 | 39.94±0.57 | 56.08±0.89 | 77.13±0.76 | 79.66±1.37 |
| EAT | ACC | 36.89±0.15 | 39.07±1.51 | 49.37±0.19 | 52.13±0.00 | 47.26±0.32 | 36.06±1.24 | 57.19±0.66 | 57.22±0.73 | 33.46±0.18 | 43.36±0.87 | 56.69±0.34 | 58.47±0.45 |
| | NMI | 05.57±0.06 | 08.83±2.54 | 32.90±0.41 | 22.48±1.21 | 23.74±0.90 | 21.46±1.80 | 33.85±0.87 | 33.47±0.34 | 13.22±0.33 | 23.93±0.45 | 33.25±0.44 | 34.79±0.25 |
| | ARI | 05.03±0.08 | 06.31±1.95 | 23.25±0.18 | 17.29±0.50 | 16.57±0.46 | 21.48±0.64 | 27.71±0.41 | 26.21±0.81 | 04.31±0.29 | 15.03±0.98 | 26.85±0.59 | 28.27±0.61 |
| | F1 | 34.72±0.16 | 33.42±3.10 | 42.95±0.04 | 52.75±0.07 | 45.54±0.40 | 31.25±0.96 | 57.09±0.94 | 57.53±0.67 | 25.02±0.21 | 42.54±0.45 | 57.26±0.28 | 58.67±0.45 |
| UAT | ACC | 52.29±0.49 | 52.25±1.91 | 33.61±0.09 | 49.31±0.15 | 52.37±0.42 | 45.38±1.15 | 56.34±1.11 | 54.76±1.42 | 48.70±0.06 | 45.38±0.58 | 56.04±0.67 | 58.49±1.07 |
| | NMI | 21.33±0.44 | 21.61±1.26 | 26.49±0.41 | 25.44±0.31 | 23.64±0.66 | 24.49±0.57 | 28.15±1.92 | 25.23±0.96 | 25.10±0.01 | 22.04±2.23 | 26.99±2.11 | 28.48±1.05 |
| | ARI | 20.50±0.51 | 21.63±1.49 | 11.87±0.23 | 16.57±0.31 | 20.39±0.70 | 21.34±0.78 | 25.52±2.09 | 19.44±1.69 | 21.76±0.01 | 14.74±1.99 | 25.22±1.96 | 27.21±1.10 |
| | F1 | 50.33±0.64 | 45.59±3.54 | 25.79±0.29 | 50.26±0.16 | 50.15±0.73 | 30.56±0.25 | 55.24±1.69 | 53.61±2.61 | 45.69±0.08 | 39.30±1.82 | 54.20±1.84 | 57.40±1.57 |

node-level and edge-level augmentation, ensuring more reliable contrastive similarity. 3) The CSADA module also plays a vital role by enhancing boundary perception through adaptive differentiation of contrastive samples. The scalability and effectiveness of the CSADA module on other CAGC methods are further validated in Appendix C.2. 4) Additionally, the dynamic confidence factor facilitates progressive learning by expanding the set of high-confidence samples, further boosting representation discriminability.

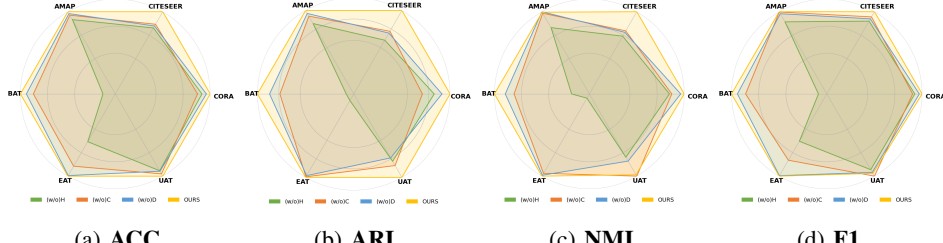

| (a) **ACC** | (b) **ARI** | (c) **NMI** | (d) **F1** |
|---|---|---|---|

Figure 3: The clustering results of three ablation variants and RAGC on the six used datasets, where the proportion between best result achieved by RAGC and each ablation variant is shown.

## 3.4 Robustness to Noise

To evaluate robustness to noise, this section presents a performance comparison between RAGC and several representative CAGC methods, including SCGC [21], DCRN [29], and CCGC [30], under various noisy conditions. To be specific, Gaussian noise $\mathcal{N}(0, \sigma_N)$ is added to the attribute features with varying standard deviations: 0.1, 0.2, and 0.3. The experimental results, presented in Table 3, reveals the following key observations. 1) The proposed RAGC consistently outperforms all the other comparisons across all metrics, even under varying levels of noise disturbances. 2) Although the performance of all methods declines as the noise ratio increases, RAGC exhibits the smallest average degradation ratio across all metrics and datasets, demonstrating its superior robustness to noise. For example, the performance of RAGC decreases by 11.36%, while DCRN experiences a 19.03% drop. The robustness of RAGC to noise is mainly attributed to its comprehensive HCA data augmentation and discriminative contrastive setting.

## 3.5 Visualization Analysis

To intuitively demonstrate the advantages of RAGC in discriminative representation learning, the raw attribute feature and embedding representations learned by several representative AGC methods are

Table 3: The performance comparison of SCGC [21], DCRN [29], CCGC [30], and RAGC with different noise levels. Negative values indicate the performance degradation percentages compared to corresponding methods without noise.

| Method | $\sigma_N$ | Cora | | | | Citeseer | | | | UAT | | | | Avg. |
|---|---|---|---|---|---|---|---|---|---|---|---|---|---|---|
| | | ACC | NMI | ARI | F1 | ACC | NMI | ARI | F1 | ACC | NMI | ARI | F1 | |
| SCGC | 0.1 | 71.58 (-3) | 53.29 (-5) | 48.72 (-6) | 64.51 (-9) | 67.15 (-5) | 39.52 (-13) | 39.81 (-14) | 59.01 (-9) | 50.60 (-11) | 20.37 (-27) | 18.09 (-27) | 46.64 (-16) | |
| | 0.2 | 67.63 (-8) | 47.91 (-15) | 42.16 (-19) | 61.68 (-13) | 55.11 (-22) | 27.52 (-39) | 23.42 (-49) | 49.48 (-24) | 50.08 (-11) | 20.08 (-28) | 16.75 (-32) | 48.34 (-13) | -20.61 |
| | 0.3 | 67.44 (-9) | 46.99 (-16) | 41.60 (-20) | 62.58 (-12) | 46.53 (-34) | 20.89 (-54) | 15.54 (-66) | 42.66 (-34) | 50.69 (-10) | 20.34 (-28) | 18.07 (-27) | 47.76 (-14) | |
| DCRN | 0.1 | 57.68 (-7) | 41.62 (-8) | 32.91 (-1) | 49.45 (-1) | 56.81 (-20) | 33.00 (-28) | 28.13 (-27) | 53.97 (-6) | 43.28 (-13) | 20.51 (-15) | 11.88 (-31) | 36.37 (-19) | |
| | 0.2 | 56.76 (-8) | 40.79 (-10) | 32.46 (-2) | 48.81 (-1) | 56.33 (-21) | 28.92 (-37) | 27.31 (-30) | 52.27 (-9) | 43.19 (-13) | 15.82 (-34) | 12.24 (-29) | 37.21 (-17) | -19.03 |
| | 0.3 | 51.22 (-17) | 34.73 (-23) | 23.56 (-29) | 44.27 (-11) | 55.52 (-22) | 26.33 (-43) | 25.39 (-35) | 50.53 (-12) | 42.77 (-14) | 15.06 (-37) | 10.49 (-38) | 37.00 (-17) | |
| CCGC | 0.1 | 70.30 (-5) | 52.91 (-6) | 46.55 (-11) | 65.37 (-8) | 63.84 (-9) | 37.85 (-15) | 34.63 (-24) | 56.88 (-9) | 52.82 (-6) | 23.85 (-15) | 18.11 (-29) | 49.46 (-10) | |
| | 0.2 | 64.78 (-12) | 49.02 (-13) | 40.78 (-22) | 59.02 (-17) | 51.81 (-26) | 27.84 (-37) | 20.50 (-55) | 44.84 (-28) | 52.44 (-7) | 22.28 (-21) | 19.75 (-23) | 49.22 (-11) | -21.36 |
| | 0.3 | 60.96 (-17) | 44.73 (-21) | 34.58 (-34) | 56.67 (-20) | 45.04 (-36) | 21.51 (-51) | 14.12 (-69) | 40.11 (-36) | 52.26 (-7) | 21.01 (-25) | 20.40 (-20) | 47.43 (-14) | |
| OURS | 0.1 | 76.84 (-2) | 57.41 (-5) | 55.68 (-7) | 75.43 (-2) | 69.19 (-3) | 41.81 (-8) | 43.06 (-7) | 61.17 (-2) | 56.09 (-4) | 25.13 (-12) | 24.51 (-10) | 54.01 (-6) | |
| | 0.2 | 73.27 (-7) | 52.80 (-13) | 49.47 (-17) | 72.41 (-6) | 65.35 (-8) | 36.68 (-19) | 37.46 (-19) | 57.68 (-7) | 55.71 (-5) | 25.76 (-10) | 25.06 (-8) | 53.19 (-7) | -11.36 |
| | 0.3 | 70.30 (-11) | 48.69 (-19) | 45.05 (-25) | 69.49 (-9) | 59.10 (-17) | 29.52 (-35) | 29.58 (-36) | 51.32 (-18) | 54.40 (-7) | 24.97 (-12) | 22.45 (-17) | 52.38 (-9) | |

visualized using t-SNE [31]. As shown in Fig. 4 and Appendix C.4, different clusters are marked with distinct numbers and colors. It can be observed that the distribution of nodes with raw attribute features appears irregular. Among all AGC methods, RAGC exhibits the strongest discriminability in embedding representation, as evidenced by nodes within the same cluster being more compact and boundaries between different clusters being relatively well-defined.

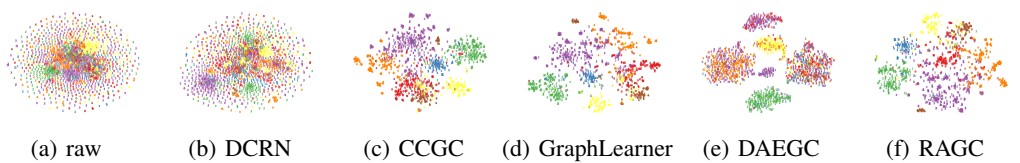

(a) raw    (b) DCRN    (c) CCGC    (d) GraphLearner    (e) DAEGC    (f) RAGC

Figure 4: The 2-D visualization on CORA dataset. The 2-D visualization of the AMAP dataset is provided in Appendix C.4.

## 4 Conclusions

In this study, a novel deep contrastive clustering method, termed RAGC, was proposed for the attributed graph clustering task, primarily comprising the HCA and CSADA modules. Specifically, RAGC takes advantage of edge-level embedding representation and performs hybrid-collaborative data augmentation by integrating both node-level and edge-level embeddings. A comprehensive contrastive similarity is constructed, which in turn provides reverse guidance for edge-level embedding augmentation. Furthermore, a novel weight modulation function-oriented contrastive strategy was designed to adaptively distinguish sample pairs according to their own characteristics, including contrastive similarity, clustering label confidence, and pseudo label correlation. This approach enhanced boundary perception and discriminative capability in self-supervised representation learning. Extensive experiments on six benchmark datasets demonstrated the promising clustering performance of RAGC and the scalability of the key CSADA module in CAGC methods. Although RAGC has demonstrated outstanding performance, there are still potential challenges when dealing with highly sparse and dynamic graphs. Exploring clustering for such graph data is a promising direction for future research.

## Acknowledgment

This research was supported by the National Natural Science Foundation of China under Grant 62403043, 62225303, and 62433004; in part by the Beijing Natural Science Foundation under Grant 4244085; in part by the Postdoctoral Fellowship Program of CPSF under Grant GZC20230203; in part by the China Postdoctoral Science Foundation under Grant 2025T180467 and 2023M740201; in part by the Interdisciplinary Research Center of Beijing University of Chemical Technology under Grant XK2025-06.

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

# A Related Studies

In recent years, deep attributed graph clustering has attracted significant attention, aiming to learn node-level embedding representation and classify nodes into disjoint clusters according to feature distribution of embedding representation. Existing AGC methods can usually be classified into three classes: generative, adversarial and contrastive [18].

## A.1 Generative and Adversarial Attributed Graph Clustering

Generative and adversarial AGC methods usually utilize Graph Convolutional Network (GCN) [32] and Graph Attention network (GAT) [33] to learn node-level embedding representation, with the core difference between them lying in their representation learning objectives. Generative AGC methods focus on recovering graph information (either attribute or structure) from the learned embedding representation [34]. For example, Deep Attention Embedded Graph Clustering (DAEGC) [35] and Parallelly Adaptive Graph Convolutional Clustering (PAGCC) [36] reconstruct the graph structure, to enhance clustering performance. On the other hand, adversarial AGC methods improve the discriminability of embedding representation by an adversarial game between a discriminator and a generator [37, 38, 39]. Furthermore, to alleviate the over-smoothing problem and obtain more comprehensive embedding representation, some methods effectively pass attribute representation through an auto-encoder to GNNs by a delivery operator in a layer-by-layer manner, such as Structural Deep Clustering Network (SDCN) [40] and Deep Fusion Clustering Network (DFCN) [41]. To unify representation learning and clustering, self-supervised training strategies are widely utilized. These strategies typically minimize the Kullback-Leibler (KL) divergence between the initial label distribution and the confidence augmented distribution, ensuring more effective clustering.

## A.2 Contrastive Attributed Graph Clustering

Although generative and adversarial AGC methods have achieved notable success, most of them rely on certain prior knowledge, such as feature distribution or label distribution, making their performance relatively sensitive to these assumptions. In contrast, contrastive learning, a self-supervised representation learning paradigm, has been rapidly extended to AGC. It constructs self-supervision signals by maximizing the similarity between the embeddings of positive samples while minimizing the similarity between the negative ones. Contrastive Multi-View Representation Learning on Graphs (MVGRL) [42] generates augmented structure by a graph diffusion network and maximizes the mutual information between cross-views through the InfoMax contrastive loss. Similarly, Self-supervised Contrastive Attributed Graph Clustering (SCAGC) [43] constructs two augmented graphs by adding noise to attribute and applying random edge perturbation to structure. It then performs contrastive learning at both the node-level and the clustering label-level. Similar to the augmentation strategies in [43], Dual Correlation Reduction Network (DCRN) [29] further reduces redundant correlation at both the feature level and the sample level to alleviate the representation collapse problem. To eliminate the need for complex and manual data augmentation, several learnable augmentation strategies have been developed. Simple Contrastive Graph Clustering (SCGC) [21] first smooths graph signal using a low-passing Laplacian filter and then utilizes parameter-unshared MLPs to generate augmented views, and further constructs a neighbor-oriented contrastive loss. Graph Node Clustering with Fully Learnable Augmentation (GraphLearner) [23] dynamically constructs an augmented graph through rich learnable structure and attribute augmenters in a self-cycle way and utilizes normalized temperature-scaled cross-entropy loss to pull positive samples closer while pushing negative samples apart.

However, most existing the methods treat all contrastive samples equally, limiting the discriminability of contrastive learning. Recently, some studies focus on hard samples, recognizing their crucial role in improving both the generalization and discriminability of the model. Structure-enhanced Heterogeneous Graph Contrastive Learning (STENCIL) [24] focuses on hard negative samples and enriches model training by randomly mixing up negative samples with highest similarity. Similarly, Xia et al. [44] establishes a more suitable measure criterion for hard negative samples through probability estimators and focuses on them by weight adjustment or a node mixing strategy. Furthermore, to mitigate interference from false negatives, Niu et. al [25] propose an affinity uncertainty-based hard negative mining approach for graph contrastive learning, which evaluates the hardness of negative samples according to collective affinity information. Beyond negative sample pairs, Liu et al. [26]

also extend the focus to hard positive samples by a unifying Hard Sample Aware Network (HSAN), strengthening the boundary perception capability.

Obviously, graph data augmentation and contrastive objective play a crucial role in contrastive AGC methods. However, most existing CAGC methods fail to fully leverage edge information. Specifically, they only utilize edge information to obtain node-level embedding representation and achieve a single augmentation. The edge-level embedding representation learning and augmentation are ignored, preventing the natural modeling of collaborative interaction across multiple levels. In addition, existing methods struggle to sufficiently perceive contrastive samples, failing to differentiate them from the perspectives of hard-positive, hard-negative, easy-positive, and easy-negative. Adaptively perceiving and significantly distinguishing these samples is crucial for improving the discriminative capability of representation learning.

# B    Experimental Setting

## B.1    Datasets

Following the key baseline in [26], the proposed RAGC is evaluated on six challenging graph benchmark datasets, including CORA [28], CITESEER (CITE) [28], AMAP [29], BAT[21], EAT [21] and UAT [21]. The detailed statistics of these datasets are briefly summarized in Table 4.

## B.2    Baselines

In the experiment, RAGC is compared against 11 state-of-the-art baseline methods, including:

- four generative and adversarial AGC methods: DAEGC [35], SDCN [40], DFCN [41] and Adversarially Regularized Graph Auto-encoder (ARGA) [37];

- four classic CAGC methods: Adaptive Graph Embedding (AGE) [28], Neighbor Contrastive Learnable Augmentation (NCLA) [45], Cluster-guided Contrastive Graph Clustering Network (CCGC) [30], GraphLearner (GL) [23];

- three hard sample aware CAGC methods: Graph Debiased Contrastive Learning with Joint Representation Clustering (GDCL) [46], ProGCL [44], HSAN [26].

Please refer to Section A or the original papers for a detailed description of these methods. For fair comparison, the original clustering results of all comparison methods are directly taken from their respective papers.

Table 4: The statistics of all used datasets and parameter settings of proposed RAGC method.

| Dataset | Statistics | | | | Parameters | | |
|---|---|---|---|---|---|---|---|
| | **Sample** | **Dimension** | **Edge** | **Class** | $\beta$ | $\gamma$ | $lr$ |
| **CORA** | 2708 | 1433 | 5429 | 7 | 0.9 | 2 | 1e-3 |
| **CITESEER** | 3327 | 3703 | 4732 | 6 | 0.9 | 1 | 1e-3 |
| **AMAP** | 7650 | 745 | 119081 | 8 | 0.1 | 2 | 5e-5 |
| **BAT** | 131 | 81 | 1038 | 4 | 0.9 | 1 | 1e-3 |
| **EAT** | 399 | 203 | 5994 | 4 | 0.7 | 5 | 1e-4 |
| **UAT** | 1190 | 239 | 13599 | 4 | 0.8 | 5 | 1e-4 |

## B.3    Parameter Setting

During the training process, the number of training epoch is set to $400$. The detailed settings of weight factors $\beta$, $\gamma$ and learning ratio $lr$ are shown in Table 4. In the HCA module, both the node-level embedding encoders and edge-level embedding encoders are both single-layer MLPs with unshared parameters. The embedding dimension is set to $1500$ for CORA, CITESEER, BAT and EAT datasets, and set to $1000$ for AMAP and UAT datasets, respectively. In the attribute augmenter, the standard deviation $\sigma_{\mathrm{N}}$ of Gaussian noise is $0.001$ and the mask ratio $r$ of the mask matrix is $0.005$.

### B.4 Metrics

To comprehensively evaluate the clustering performance of all AGC methods, four widely used clustering metrics are utilized, including Accuracy (ACC), Normalized Mutual Information (NMI), Average Rand Index (ARI) and macro F1-score (F1). Each of these metrics is positively correlated with clustering performance. All methods are conducted ten times and the average values along with their corresponding standard deviations of all clustering metrics are reported.

### B.5 Computing Resource Details

The experimental environment is a server equipped with an Intel(R) Xeon(R) Gold 6348 CPU, a NVIDIA A800 PCIe 80GB GPU, 100GB RAM, and the PyTorch deep learning platform. The training time per run for each dataset is less than 3 minutes.

## C Additional Experiments

In this section, additional experiments are conducted to further verify the robustness and effectiveness of the proposed RAGC. These include a detailed analysis of the performance comparison experiment (Section C.1), an investigation of the scalability of the CSADA module (Section C.2), sensitivity analysis of parameters $\beta$, $\gamma$, $\sigma_{\mathrm{N}}$, and $r$ (Section C.3), as well as comprehensive visualization results (Section C.4).

### C.1 Performance Comparison

The quantitative experimental results of the all AGC comparative methods on the benchmark datasets are shown in Table 2. Notably, except for a few cases, the proposed RAGC achieves optimal clustering performance across all six datasets, clearly demonstrating its superiority and effectiveness. Specifically, the performance improvements of RAGC are significant on several datasets. For example, on the BAT dataset, RAGC outperforms the sub-optimal HSAN method by 2.65%, 2.53%, 2.44% and 2.53% in terms of ACC, NMI, ARI and F1 metrics, respectively. Similarly, for the CORA dataset, the RAGC exceeds the sub-optimal HSAN by 1.67%, 1.41%, 2.32% and 1.80% on ACC, NMI, ARI and F1 metrics, respectively.

Further, several fine-grained analyses are drawn from the following aspects:

1) It is clear that RAGC always achieves better performance on all six datasets compared to the classical generative-based and adversarial-based AGC methods. For example, RAGC achieves an average improvement of 17.28%, 9.13%, 15.78% and 21.08% over DFCN in terms of ACC, NMI, ARI and F1, respectively. The primary reason is that classical generative and adversarial based AGC methods learn node-level embedding representation under the guidance of prior knowledge, making their discriminability sensitive to these assumptions. They lack specialized contrastive strategies to achieve information alignment and fully leverage self-supervised information from the discriminative latent space, leading to poor clustering performance. In most cases, contrastive AGC methods outperform classical generative-based and adversarial-based AGC methods, further validating the effectiveness of the contrastive learning strategy.

2) Among all contrastive AGC methods, RAGC remains superior, with its performance advantage being even more pronounced. As an example, on the CORA dataset, RAGC outperforms the CCGC method by 4.86%, 4.17%, 7.33% and 5.93% in terms of ACC, NMI, ARI and F1, respectively. The main reason for this superior performance is that RAGC enhances the discriminability of embedding representation effectively by distinguishing contrastive sample pairs, with a particular emphasis on hard samples. In addition, hard sample aware contrastive AGC methods (such as GDCL, ProCGL, and HSAN) achieve better performance on some datasets. However, they still underperform compared to the proposed RAGC on all datasets, demonstrating that RAGC is more effective in learning discriminative embedding representation. Especially compared to the important baseline HSAN, on all datasets, RAGC achieves an average improvement of 1.66%, 1.25%, 1.53% and 1.51% across all datasets in terms of ACC, NMI, ARI, and F1, respectively. This superior performance is primarily due to RAGC's hybrid-collaborative augmentation, which integrates both node-level embedding representation and edge-level embedding representation, resulting in a more comprehensive and discriminative contrastive similarity. In addition, the designed contrastive sample adaptive-differential

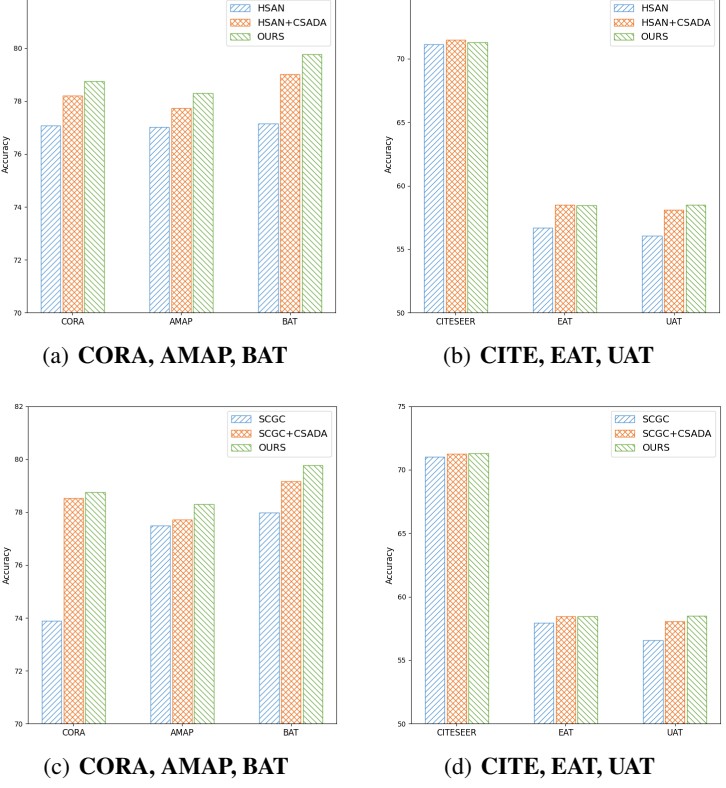

(a) **CORA, AMAP, BAT**     (b) **CITE, EAT, UAT**

(c) **CORA, AMAP, BAT**     (d) **CITE, EAT, UAT**

Figure 5: The experimental results of HSAN [26] and SCGC [21] with the proposed CSADA module on all datasets.

awareness strategy allows RAGC to effectively distinguish sample pairs, especially positive-hard, positive-easy, negative-hard, and negative-easy ones. This significantly enhances its boundary perception ability in representation learning, leading to improved clustering performance.

In conclusion, these experimental results and analyses evidently validate the effectiveness of the proposed RAGC method in attributed graph clustering task.

## C.2 Scalability of the CSADA module

To further validate the scalability and compatibility of the proposed CSADA strategy, it is transferred to other contrastive DGC methods, including SCGC [21] and HSAN [26]. Specifically, the data augmentation strategy and similarity construction across augmented views remain with the original settings in [26, 21], and only the contrastive setting is replaced by the proposed CSADA strategy. As shown in Fig. 5, in most cases, the clustering performance of these methods significantly improves when combined with the proposed CSADA strategy. This demonstrates that CSADA is not only compatible with existing CAGC methods but also exhibits strong scalability.

## C.3 Parameter Sensitivity Analysis

Several key parameters influence the performance of the RACG method, including the weight factors $\beta$, $\gamma$ in the CSADA module, as well as the noise standard deviation $\sigma_N$ and mask ratio $r$ in the HCA module. The performance sensitivity analysis of RAGC with respect to these parameters are examined as follows.

### C.3.1 Sensitivity Analysis of $\beta$ and $\gamma$

The factors $\beta$ and $\gamma$ control the weight magnitude of contrastive samples. The search ranges of these weight factors $\beta$ and $\gamma$ are $\{0.1, 0.3, 0.5, 0.7, 0.9\}$ and $\{1, 2, 3, 4, 5\}$, respectively. As shown in

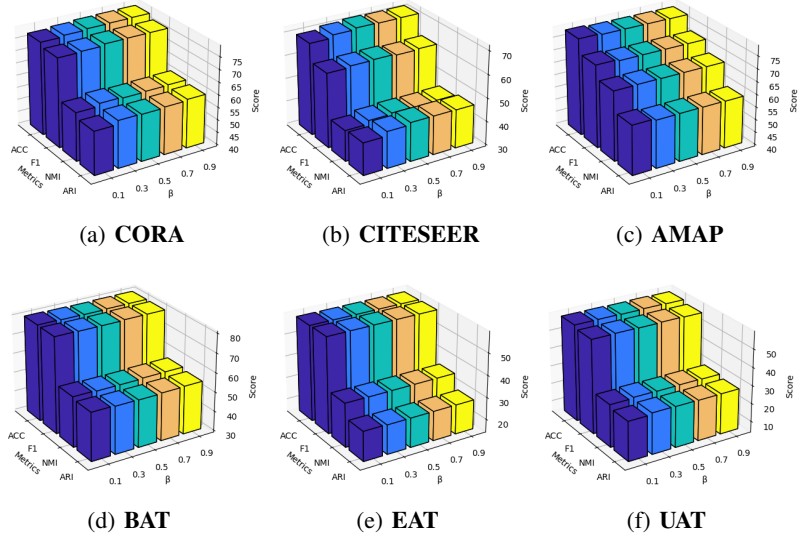

Figure 6: The clustering results of RAGC with different $\beta$ on six datasets.

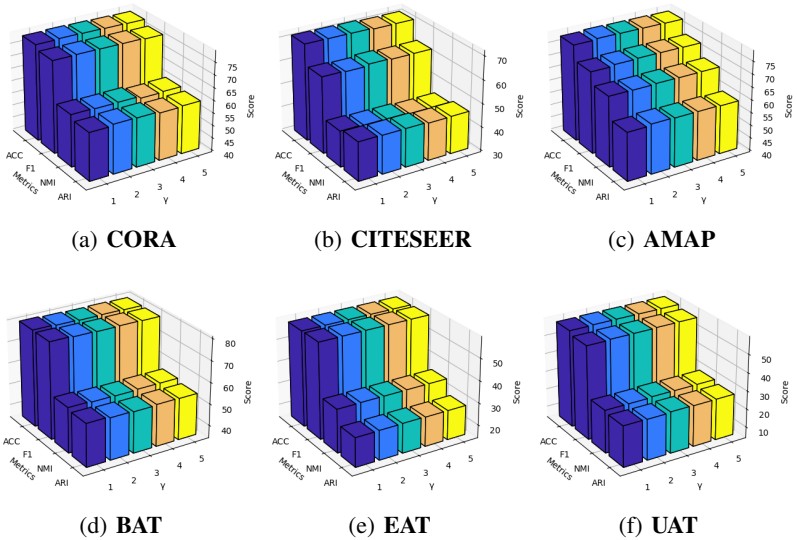

Figure 7: The clustering results of RAGC with different $\gamma$ on six datasets.

Fig. 6 and Fig. 7, all clustering metrics across different datasets exhibit commendable performance and excellent stability when $\beta$ and $\gamma$ across these parameter variations. The experimental results show that the proposed RAGC is insensitive to $\beta$ and $\gamma$, demonstrating its strong robustness to these hyperparameters.

### C.3.2 Sensitivity Analysis of parameters $\sigma_N$ and $r$

The candidate ranges for noise standard deviation and mask ratio in the HCA module are $\sigma_N \in \{0.0001, 0.001, 0.01, 0.1, 1\}$ and $r \in \{0.0005, 0.005, 0.05, 0.5, 0.7, 0.8\}$, respectively. Fig. 8 and Fig. 9 show the variations in ACC metrics with respect to $\sigma_N$ and $r$. It can be observed that the performance of RAGC remains relatively stable when $\sigma_N$ varies in relatively wide range $\{0.0001, 0.001, 0.01, 0.1\}$, indicating its robustness to moderate noise perturbations. Despite noticeable performance fluctuations when $r$ varies within the given range, RAGC still maintains relatively excellent performance when $r \in \{0.0005, 0.005, 0.05\}$. This suggests that moderate disturbance (such as noise or feature mask) can enhance the generalization ability and robustness of representation learning. However, excessive noise and feature masks may cause semantic drift in attribute

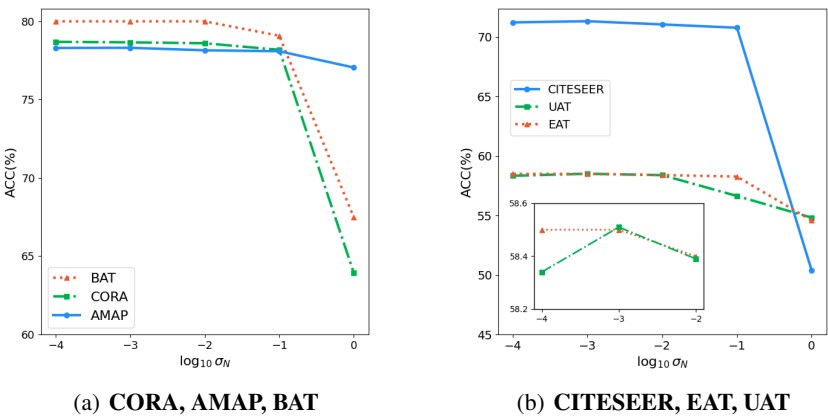

(a) **CORA, AMAP, BAT**

(b) **CITESEER, EAT, UAT**

Figure 8: The sensitivity analysis of standard deviation $\sigma_N$ in Gaussian noise.

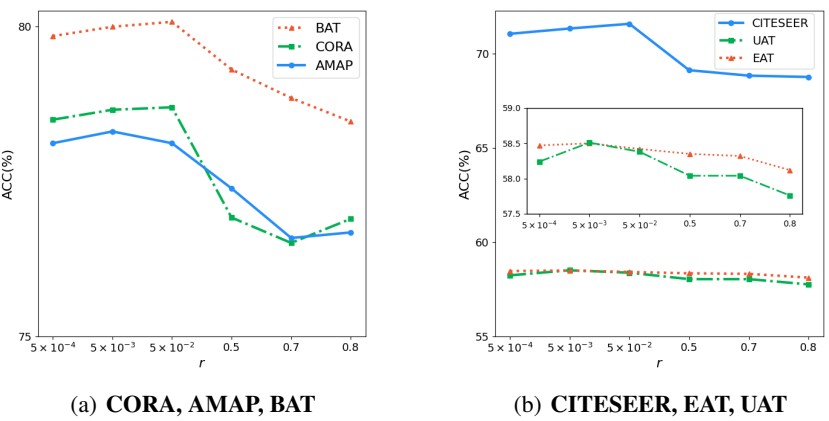

(a) **CORA, AMAP, BAT**

(b) **CITESEER, EAT, UAT**

Figure 9: The sensitivity analysis of the mask ratio $r$ in attribute mask matrix.

information, increasing the difficulty of representation learning, which in turn hinders discrimination and degrades clustering performance.

## C.4 Visualization

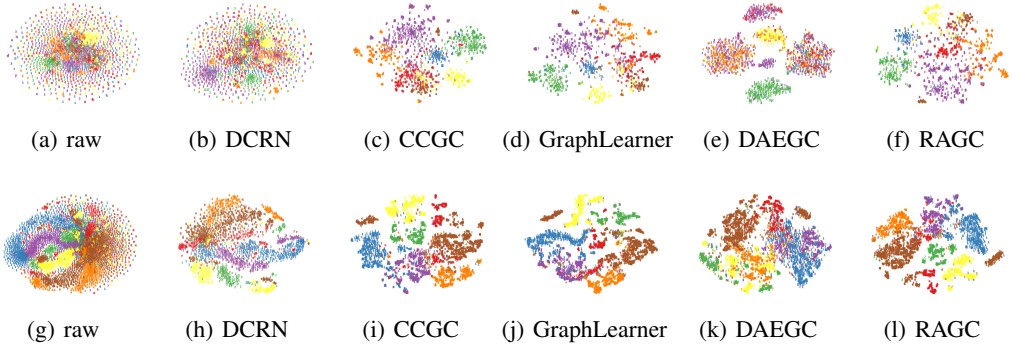

(a) raw    (b) DCRN    (c) CCGC    (d) GraphLearner    (e) DAEGC    (f) RAGC

(g) raw    (h) DCRN    (i) CCGC    (j) GraphLearner    (k) DAEGC    (l) RAGC

Figure 10: The 2-D visualization on two datasets. The first row and second row correspond to CORA and AMAP datasets, respectively.

