# OpenReview forum: "Hybrid-Collaborative Augmentation and Contrastive Sample Adaptive-Differential Awareness for Robust Attributed Graph Clustering"
_NeurIPS.cc/2025/Conference — NeurIPS 2025 poster_

### Official Review · Reviewer_PbCi · 2025-06-26

**Clarity:** 4
**Significance:** 3
**Originality:** 3
**Rating:** 5
**Confidence:** 4

**Summary:**

In the paper, the authors propose an unsupervised graph clustering method RAGC to improve the clustering performance, through combining two modules Hybrid-Collaborative Augmentation (HCA) and Contrastive Sample Adaptive-Differential Awareness (CSADA). The HCA module combines the dual collaborative augmentation of node-level and edge-level, and the CSADA module utilizes some dynamic prior information to differentiate the importance of contrastive sample pairs with an elaborate weight function from four key perspectives, eventually improving the discriminability of self-supervised graph embedding representation learning. The experimental results show significantly performance than existing methods on benchmark datasets. The theoretical analysis is sound and experimental verification is convining.

**Questions:**

Q1: The CSADA module is dependent on the accuracy of pseudo-labeling in the contrastive sample weighting process, but the paper lacks a systematic analysis of the impact of pseudo-labeling errors on model performance. It is recommended to supplement the relevant analysis so as to enhance the completeness and persuasiveness of the paper.

Q2. The dynamic pseudo-label selection mechanism is one of the innovations of this study, but how exactly does it dynamically change and converge during the training process?

Q3. Sensitivity experiments on the weight adjustment parameters β and γ show that they are robust over a range, but how are the parameter ranges chosen? Please explain the selection principle, or provide a theoretical or empirical evidence.

**Ethical Concerns:**

["NO or VERY MINOR ethics concerns only"]

**Final Justification:**

Following the rebuttal period and the response from the authors, I believe my rating remains appropriate and will keep it as is.

**Limitations:**

Yes

**Paper Formatting Concerns:**

I did not find any major formatting issues.

**Quality:**

3

**Strengths And Weaknesses:**

The strengths:
S1. To my knowledge, this is the first work to systematically categorize and distinguish the contrastive samples from four key criteria.

S2. The work enhances the granularity and effectiveness of contrastive representation learning, and the  design is reasonable and interesting for general contrastive learning.

S3. The experiments are convincing. The overall quality of the paper is high and has potential for acceptance.

The weakness:
W1. This paper shows a relatively good readability, but there are still some writing and layout issues.

W2. The paper does not provide a detailed analysis of the impact of the quality of pseudo-label information on the performance of RAGC.

---

> ### Author Rebuttal · Authors · 2025-07-29
>
> **Reply**: Thank you for your thoughtful and detailed review. We appreciate your recognition of our contributions and the advantages of the proposed method. We will carefully address your issue below.
>
> **Q1**: *The CSADA module is dependent on the accuracy of pseudo-labeling in the contrastive sample weighting process, but the paper lacks a systematic analysis of the impact of pseudo-labeling errors on model performance. It is recommended to supplement the relevant analysis so as to enhance the completeness and persuasiveness of the paper.*
>
> **Reply**: Thank you for your valuable suggestion. To mitigate the potential impact of pseudo-labeling errors on contrastive sample weighting, we designed several specific mechanisms into the RAGC model. On one hand, the CSADA module adopts a dynamic confidence-based weighting contrastive sample selection strategy, where only a small number of high-confidence samples are used for weighting during the early stages of training, thereby reducing the influence of inaccurate pseudo-labels on contrastive learning. On the other hand, as training progresses, pseudo labels are dynamically updated and gradually corrected, which improves the overall robustness of the model. Additionally, to further assess the impact of pseudo-label errors, we conducted 10 repeated experiments and reported the mean and standard deviation of clustering metrics in the comparison study (see Table 1). The results show minimal performance fluctuations, demonstrating the stability of our mechanism under different pseudo-label initialization conditions. In addition, the noise disturbance of the graph will reduce the quality of pseudo labels. We tested the robustness of proposed RAGC method to noise under different noise conditions (Gaussian noise $\mathcal N(0, \sigma_N)$ with $ \sigma_N \in $ {0.1, 0.2, 0.3} ), as shown in Section 3.4 and Table 2. The following key observations could be concluded: 1) The proposed RAGC consistently outperforms all the other comparisons across all metrics, even under varying levels of noise disturbances. 2) Although the performance of all methods declines as the noise ratio increases, RAGC exhibits the smallest average degradation ratio across all metrics and datasets, demonstrating its superior robustness to noise. For example, the performance of RAGC decreases by 11.36\%, while DCRN experiences a 19.03\% drop. This indirectly verifies the robustness of our method to pseudo labels.
>
> We sincerely appreciate your suggestion and will consider adding further analysis and discussion to enhance the completeness and persuasiveness of the paper.
>
> **Q2**: *The dynamic pseudo-label selection mechanism is one of the innovations of this study, but how exactly does it dynamically change and converge during the training process?*
>
> **Reply**: Thank you for your question. Our dynamic pseudo-label selection mechanism is implemented through a gradually decreasing confidence threshold $\tau$ combined with the iterative update of pseudo-labels. In the early stages of training, a high value of $\tau$ is used to select only a small set of high-confidence sample pairs for weighted contrastive learning, thereby reducing the impact of pseudo-label noise. As training progresses, $\tau$ is progressively reduced following the update rule $\tau = \max(\tau \times 0.1, \tau \times 0.95)$, and eventually stabilizes at $\tau \times 0.1$. This allows the coverage of high-confidence samples to gradually expand over training. Meanwhile, pseudo-labels are dynamically updated in each training epoch using K-means clustering, ensuring consistency with the current node representations and improving both representation quality and the reliability of contrastive learning.
>
> **Q3**: *Sensitivity experiments on the weight adjustment parameters $\beta$ and $\gamma$ show that they are robust over a range, but how are the parameter ranges chosen? Please explain the selection principle, or provide a theoretical or empirical evidence.*
>
> **Reply**: Thank you for your question. The parameter ranges $\beta \in (0,1)$ and $\gamma \in [1,5]$ are carefully chosen to enable more discriminative weighting of the four types of contrastive sample pairs in the CSADA module. As illustrated in Fig. 2 and Eq. (18), the weighting functions are designed to dynamically adjust sample weights based on their similarity scores. Specifically, $\beta$ controls the weight differentiation between hard and easy positive pairs, i.e., hard positive pairs with lower similarity receive increased weights, while easy positive pairs with higher similarity still maintain relatively high weights. $\gamma$, on the other hand, governs the negative sample weighting, i.e., hard negative pairs with high similarity are down-weighted to ensure they are effectively pushed apart in the embedding space. As shown in Fig. 2, the current parameter ranges effectively achieve this intended behavior. Values outside these ranges tend to result in indistinguishable weighting across samples, thereby undermining the purpose of sample differentiation. Additionally, sensitivity analysis experiments conducted across various values confirm that the selected intervals for $\beta$ and $\gamma$ offer both robustness and adaptability, ensuring stable performance of the model.

---

> > ### Comment · Reviewer_PbCi · 2025-08-04
> >
> > Thanks for the reply. I now have a better understanding of the paper and will keep my positive score.

---

> > > ### Author Response · Authors · 2025-08-06
> > >
> > > We truly appreciate the time and effort you invested in reviewing our work.

---

> ### Author Response · Authors · 2025-08-02
>
> We hope our rebuttal can address your concerns. If you have further questions, we are willing to reply. Thank you!

---

### Official Review · Reviewer_uEFh · 2025-06-29

**Clarity:** 3
**Significance:** 4
**Originality:** 4
**Rating:** 5
**Confidence:** 4

**Summary:**

The paper develops a RAGC framework to systematically improve representation discrimination in the contrastive graph clustering task by combining Hybrid-Collaborative Augmentation and Contrastive Sample Adaptive-Differential Awareness. The HCA module augments both node-level and edge-level features to achieve multi-granularity information fusion. The CSADA module dynamically adjusts the importance of different contrastive samples according to confidence level of pseudo-label and semantic similarity, thereby realizing differential learning between positive, negative, easy, and hard samples. Experiments verify the superior performance of RAGC on multiple datasets.

**Questions:**

1. In the HCA module of the model, why was the parameter-unshared MLP chosen to implement the data augmentation? If it is an empirical choice, it needs to be explained. Are there other, more efficient structures for enhancing the representation of graph data?
2. The experiments were limited to homogeneous graph data (e.g., citation networks), and the generalization ability of the method on heterogeneous should be verified.
3. In the HCA module, how do the node-level embedding augmenter and edge-level embedding augmenter collaborate?
4. The CSADA module relies on pseudo-labels for sample weighting, but do different pseudo-label initialization strategies have an impact on the final clustering performance? The authors are requested to further explain the specific methodology and selection basis of the current pseudo-label initialization, and to indicate whether the process has better stability under various initialization.

**Ethical Concerns:**

["NO or VERY MINOR ethics concerns only"]

**Final Justification:**

Most of my concerns have been addressed, and I have updated my scoring accordingly. And,  I recommend acceptance of this paper.

**Limitations:**

Yes

**Paper Formatting Concerns:**

I have carefully checked the manuscript for compliance with the NeurIPS 2025 formatting instructions and did not observe any significant formatting problems.

**Quality:**

3

**Strengths And Weaknesses:**

The study proposes targeted improvements in both node-edge dual augmentation and contrastive sample differential perception, with good innovations and technical contributions. The experimental part is fully designed and validated, and the corresponding results are convincing over several representative methods. In addition, this paper also verifies the generalization of key module in other graph contrastive clustering methods, which is very interesting and valuable. More details of existing weakness can be found in the Questions section.

---

> ### Author Rebuttal · Authors · 2025-07-29
>
> **Reply**: Thank you for your thoughtful and detailed review. We appreciate your recognition of our contributions and the advantages of the proposed method. We will carefully address your concerned issues below.
>
> **Q1**: *In the HCA module of the model, why was the parameter-unshared MLP chosen to implement the data augmentation? If it is an empirical choice, it needs to be explained. Are there other, more efficient structures for enhancing the representation of graph data?*
>
> **Reply**: Thank you for your question. The MLP module performs simple nonlinear transformations on the low-pass filtering representations to generate diversified augmented views. Specifically, we design un-shared MLPs as augmenters to learn different contrastive representations, thereby achieving the effects of traditional data augmentation methods (e.g., attribute perturbation or structural modification) in a more flexible manner. This implicit augmentation approach avoids the complexity of directly modifying node attributes or graph structures, while preserving semantic consistency. Moreover, as a pure feature transformer, the MLP operates independently of the graph structure, which decouples it from the graph convolution operator and avoids the intricate entanglement often seen in conventional graph augmentation methods. This enhances the overall flexibility and generalizability of the framework. While other methods, such as Graph Attention Networks (GAT), could in theory also capture structural information, they typically require additional neighborhood aggregation or parameterized augmentation strategies, introducing more computational complexity. In contrast, the generality of MLP makes it suitable for various types of graph data, such as citation networks and traffic networks.
>
> In summary, the use of MLP achieves a favorable balance among efficiency, generalization, and performance, making it a theoretically sound and practically effective design choice.
>
> **Q2**: *The experiments were limited to homogeneous graph data (e.g., citation networks), and the generalization ability of the method on heterogeneous should be verified.*
>
> **Reply**: Thank you for your valuable suggestion. Although our method does not explicit design for heterogeneous node or edge issue, the overall design of our framework is inherently adaptable to heterogeneous graphs. In particular, the HCA module performs node-level and edge-level augmentation through a unified similarity fusion mechanism that does not depend on node or edge homogeneity, allowing it to naturally accommodate varying semantic and structural relationships across different types. Similarly, the CSADA module conducts confidence-aware contrastive weighting without assuming type-specific information, making the entire contrastive learning process type-agnostic and structurally flexible. These architectural properties provide a solid theoretical foundation for the potential generalization to heterogeneous graphs.
>
> In experiment, we have evaluated the proposed method on datasets such as BAT and EAT, involving non-homophilous structural characteristics. From a semantic perspective, heterophily graphs can be viewed as a special type of heterogeneous graphs, where the heterogeneity stems from label interactions rather than node or edge types. Compared with the representative HSAN method, our proposed RAGC achieves average improvements on the BAT and EAT datasets of $2.20$% in ACC, $2.04$% in NMI, $1.93$% in ARI, and $1.97$% in F1. The excellent performance achieved by our model on them provides supporting evidence for its potential to generalize to broader heterogeneous graph scenarios. We thank you again for raising this valuable point. In the future, we will attempt to extend the proposed RAGC to general heterogeneous graph clustering.
>
>
> **Q3**: *In the HCA module, how do the node-level embedding augmenter and edge-level embedding augmenter collaborate?*
>
> **Reply**: Thank you for your question. In the HCA module, the node-level and edge-level augmenters collaborate through a joint similarity fusion mechanism. Specifically, we first perform separate augmentations on node attributes and graph structure to obtain node-level embeddings ($\mathbf Z^a$, $\mathbf Z^b$) and edge-level embeddings ($\mathbf E^a$, $\mathbf E^b$). These embeddings are then integrated within a unified contrastive framework by constructing a fused similarity matrix $\mathbf S^{l,m} = \alpha \mathbf Z^l \left(\mathbf Z^m\right)^T+(1-\alpha)\mathbf E^l \left(\mathbf E^m\right)^T$, which allows both node-level and edge-level information to contribute to a shared contrastive objective. Furthermore, we dynamically update the edge structure representations using attribute information (as described in Eqs. (11)–(12)), enabling the edge-level augmenter to receive feedback from node-level embeddings during each training iteration. This facilitates mutual interaction and structural refinement. In this framework, the node-level augmenter provides semantic guidance, while the edge-level augmenter imposes structural constraints. They work collaboratively and promote each other in the representation space, effectively enhancing the discriminative capability.
>
> **Q4**: *The CSADA module relies on pseudo-labels for sample weighting, but do different pseudo-label initialization strategies have an impact on the final clustering performance? The authors are requested to further explain the specific methodology and selection basis of the current pseudo-label initialization, and to indicate whether the process has better stability under various initialization.*
>
> **Reply**: Thank you for your valuable suggestion. In this study, we chose K-means clustering as the initialization method for pseudo-labels mainly because it is widely used in unsupervised scenarios, easy to implement, and computationally efficient. Compared to other clustering algorithms such as spectral clustering or DBSCAN, K-means is more lightweight and easier to integrate with the graph neural network training process. Additionally, in the CSADA module, we adopt a dynamic confidence mechanism, where only a small number of high-confidence samples are selected for weighted contrastive learning during the early training phase. This helps to mitigate the impact of noisy pseudo-labels at initialization. To further validate the stability of this approach, we performed 10 independent runs on all datasets and reported the mean and standard deviation of clustering metrics (see Table 1). The results demonstrate that our model maintains stable performance under different K-means initializations, with only minor fluctuations.
>
> In summary, we believe that K-means offers advantages in terms of efficiency, stability, and ease of use, making it a suitable pseudo-labeling strategy for our framework.

---

> > ### Comment · Reviewer_uEFh · 2025-08-05
> >
> > Thank you for your detailed response for my concerns. Based on the above, I recognize the innovation of this paper. Most concerns are addressed, especially in terms of collaborative augmentation between node-level embedding augmenter and edge-level embedding augmenter and  details of CSADA module. The scalability of heterogeneous graphs is a direction worthy of special research in attributed graph clustering. The authors have provided them own insights, but further consideration is still needed in future work.

---

> > > ### Author Response · Authors · 2025-08-06
> > >
> > > Thank you for acknowledging my response. Indeed, heterogeneous graph clustering is a challenging research area, and our method design has certain applicability to it. This study primarily focused on data augmentation and contrastive settings in general contrastive graph learning. In the future, heterogeneous graph contrastive clustering will be our primary focus. For instance, we can decompose a heterogeneous graph into several homogeneous graphs and conduct separate contrastive learning on each homogeneous graph.

---

### Official Review · Reviewer_ryRm · 2025-07-01

**Clarity:** 3
**Significance:** 3
**Originality:** 4
**Rating:** 5
**Confidence:** 5

**Summary:**

This study proposes a novel graph contrastive clustering framework, RAGC, consisting of two crucial modules, HCA and CSADA. This proposed RAGC method simultaneously utilizes node embedding representation and edge embedding representation for comprehensive contrastive learning, and distinguishes different types of positive and negative sample pairs through an adaptive weighting mechanism, thus enhancing the discriminability and robustness of representation learning and clustering results. The experimental settings are sufficiently reasonable and it shows better performance than existing mainstream methods.

**Questions:**

Q1: The numerical significance of Fig.3 is unclear when expressing the ablation results.

Q2: The dynamic confidence selection mechanism gradually expands the set of high-confidence samples during the training process, is this strategy still applicable when the sample distribution is extremely unbalanced at the beginning of training or when there is a high proportion of noisy samples?

Q3: In CSADA module, confidence parameter $\tau$ is adaptive and varies with the  training epoch. What are the advantages of this setting?

Q4: Why did CSADA use positive/negative and easy/hard divisions rather than finer grained or other division criteria when it differentiated the sample into four categories for weighting?

**Ethical Concerns:**

["NO or VERY MINOR ethics concerns only"]

**Final Justification:**

All my concerns are addressed. Thus, I keep my positive score.

**Limitations:**

Yes

**Paper Formatting Concerns:**

The manuscript generally follows the NeurIPS 2025 formatting guidelines, and I did not notice any major issues in this regard.

**Quality:**

4

**Strengths And Weaknesses:**

Strengths

S1: The study introduces edge-level augmentation to fully utilize the structural information with a complementary and collaborative way, which aims at improving the completeness of representation learning and contrastive learning;

S2: The sample weighting mechanism takes into account the differences between positive and negative samples as well as easy and difficult samples, which significantly improves the contrastiveness of self-supervised representation learning;

S3: The CSADA module has strong generalizability and can be potentially integrated into other contrastive learning frameworks with migration and extension value.

Weaknesses

W1: The scalability of this method for large-scale data needs further verification.

---

> ### Author Rebuttal · Authors · 2025-07-29
>
> **Reply**: Thank you for your thoughtful and detailed review. We appreciate your recognition of our contributions and the advantages of the proposed method. We will carefully address your issue below.
>
> **W1**: *The scalability of this method for large-scale data needs further verification.*
>
> **Reply**: Thank you for pointing out this important issue. We acknowledge that the current experiment only be conducted on medium-scale datasets, primarily due to limitations in our experimental hardware resources. However, the scalability has been fully considered in the model design. Specifically, we employ the low-frequency filtering combined with lightweight MLP-based augmentation, which avoids complex and computationally intensive operations, thereby effectively reducing computational complexity. Additionally, the K-means based pseudo-label generation process in our method is highly parallelizable, facilitating adaptation to large-scale data processing. Once again, we sincerely appreciate your review and valuable feedback.
>
> **Q1**: *The numerical significance of Fig.3 is unclear when expressing the ablation results.*
>
> **Reply**: Thank you for pointing out the issue regarding the clarity of the numerical physical meaning in Fig. 3. In this figure, we adopted a radar chart format to intuitively illustrate the relative performance distribution of RAGC and its three ablatiton variants (i.e., **(w/o) D**, **(w/o) H**, **(w/o) C**) across multiple clustering metrics (ACC, NMI, ARI, F1) on six datasets, where D: Dynamic High Confidence Samples Selection, H: HCA module, C: CSADA module. The radar chart reflects the proportion of the optimal result accounted for by each ablation variant and complete RAGC across different metrics, thereby highlighting the individual contributions and synergistic effects of each component. It is worth noting that since radar charts are primarily used for trend analysis and comparative visualization, the exact numerical values are not explicitly annotated in the figure. We will consider adding annotations or supplementary explanations in the ablation study section to improve the readability and rigor of the figure. Thank you again for your helpful suggestion.
>
> **Q2**: *The dynamic confidence selection mechanism gradually expands the set of high-confidence samples during the training process, is this strategy still applicable when the sample distribution is extremely unbalanced at the beginning of training or when there is a high proportion of noisy samples?*
>
> **Reply**: Thank you for your attention to the novel dynamic sample selection mechanism. In RAGC, we adopt a progressive strategy that starts with a small set of high-confidence samples. Specifically, in the early training stage, a relatively large $\tau$ value is used to select only a few top-confidence samples for hard-aware contrastive learning, ensuring the stability and reliability of early training. As the representation learning capability improves, the threshold gradually decreases, allowing more confident samples to be involved and thereby enhancing generalization. This design helps to alleviate the adverse effects of sample distribution imbalance or noisy labels at the initial stage. Additionally, in our contrastive framework, low-confidence samples are assigned a fixed weight of $W = 1$, which further reduces the risk of noise-induced misleading updating. As shown in Section 3.4, the proposed RAGC demonstrates strong robustness in the presence of noise. This robustness is largely attributed to the joint effect of the dynamic sample selection strategy and the contrastive sample weighting mechanism, together mitigating issues caused by distribution imbalance and label noise in the early training phase.
>
> Nevertheless, we acknowledge that this strategy may still face challenges when confronted with extremely poor data quality, and thus we would consider this important direction for future research.
>
> **Q3**: *In CSADA module, confidence parameter $\tau$ is adaptive and varies with the training epoch. What are the advantages of this setting?*
>
> **Reply**: Thank you for your questions. In the CSADA module, the design of gradually decreasing the confidence threshold $\tau$ over training epochs offers several advantages. First, by setting a larger $\tau$ value in the early stages of training, only a small number of high-confidence samples are selected, which helps mitigate the negative impact of inaccurate pseudo-labels and improves the stability and robustness of early training. Second, as the representation ability of model improves, $\tau$ is gradually reduced, leading to an expansion of the high-confidence sample set. This forms a progressive learning process that moves from easy to hard and from few to many, thereby enhancing the ability to distinguish complex samples. At the same time, incorporating more samples into training allows the model to capture richer semantic and structural information, further improving effectiveness of contrastive learning. Finally, this mechanism works synergistically with the weighting adjustment function, enabling adaptive training dynamics at different stages and enhancing both the representation learning capacity and generalization performance.
>
> Therefore, the dynamic sample selection mechanism not only provides a stable and progressive learning path, but also plays a key role in improving model performance and adapting to complex data structures.
>
> **Q4**: *Why did CSADA use positive/negative and easy/hard divisions rather than finer grained or other division criteria when it differentiated the sample into four categories for weighting?*
>
> **Reply**: Thank you for your question. In the CSADA module, we adopt a four-category weighting strategy based on the positive/negative and easy/hard distinction, which is motivated by both the core objectives of contrastive learning and practical feasibility. On the one hand, the positive or negative relationship between samples determines whether they should be pulled together or pushed apart in the representation space, which is fundamental to contrastive learning. On the other hand, the easy or hard level reflects the difficulty of representation learning from a particular sample pair. Combining these two aspects allows us to identify four representative categories of contrastive pairs that effectively cover the most influential training cases for the model. Compared to existing hard-sample aware schemes, our approach maintains strong expressiveness while achieving better stability and generalization, and reduces the risk of overfitting or being misled by noisy pseudo-labels. Experimental results across multiple datasets also confirm the effectiveness of this contrastive design and its scalability on other graph contrastive clustering methods. Therefore, we believe that this positive/negative and easy/hard four-type categorization sufficiently captures the key contrastive patterns and serves as an efficient and reliable choice for graph contrastive learning.

---

> > ### Comment · Reviewer_ryRm · 2025-08-05
> >
> > Thanks for the reply. All my concerns are addressed. Thus, I will keep my positive score.

---

> > > ### Author Response · Authors · 2025-08-06
> > >
> > > We sincerely thank you for you constructive feedback and for retaining a positive evaluation of the paper.

---

### Official Review · Reviewer_zM6C · 2025-07-03

**Clarity:** 3
**Significance:** 4
**Originality:** 4
**Rating:** 4
**Confidence:** 4

**Summary:**

The paper explores the robust attributed graph clustering and discusses two problems faced by existing works, i.e., insufficient data augmentation and rough differential perception of contrastive sample pairs in contrastive learning. To tackle these issues, a novel robust attributed graph clustering (RAGC), incorporating hybrid12 collaborative augmentation (HCA) and contrastive sample adaptive-differential awareness (CSADA), is proposed in this work, which integrates CSADA and HCA for better feature extraction capacities. The paper indicates its novelty and signifcance, which is also demontrated via a series of experiments.

**Questions:**

1. How about the adaptability of the work to heterogeneous graphs? I recommend to discuss this issue in brief.
2. The augmentation strategies for nodes and edges are from different feature sources, which might result in some problems of conflicting objectives. Please have a discussion on this issue.

**Ethical Concerns:**

["NO or VERY MINOR ethics concerns only"]

**Final Justification:**

This paper proposes a novel contrastive attributed graph clustering, RAGC, which mainly focuses on the issues in effective data augmentation and contrastive objective setting. This work exhibits relatively good innovation, particularly in the proposed contrastive sample adaptive-differential awareness. Although, large-scale scalability should be further considered, which is important for graph clustering. Therefore, I provide the above recommended score.

**Limitations:**

Yes

**Paper Formatting Concerns:**

No obvious formatting issues.

**Quality:**

4

**Strengths And Weaknesses:**

## Pros
1. The work is overall well-written with clear problem statement and motivations. The method is well illustrated.
2. The method design is technically solid, by employing CSADA as a reasonable and fine-grained extensions in differentiation awareness between positive and negative sample pairs.
3. The experiments are sufficient.

## Cons
1. Some details to be further explained. For example, the reason for adopting exponential weighting strategies.

---

> ### Author Rebuttal · Authors · 2025-07-29
>
> **Reply**: Thank you for your thoughtful and detailed review. We appreciate your recognition of our contributions and the advantages of the proposed method. We will carefully address your issues below.
>
> **Cons1**: *The reason for adopting exponential weighting strategies should be further explained.*
>
> **Reply**: Thank you for your insightful comment. In our CSADA module, we adopt an exponential weighting strategy to enhance the contrastiveness difference among the four types of contrastive sample pairs. This design is motivated by two key considerations:
>
> - First, the exponential functions allow the weights of high-confidence samples to grow beyond the fixed baseline weight setting ($W=1$) assigned to low-confidence pairs. This ensures that truly informative samples significantly contribute to the contrastive objective, effectively amplifying reliable signals and suppressing the influence of noisy or uncertain examples.
>
> - Second, the exponential form offers a smooth yet discriminative weighting mechanism based on sample similarity, enabling adaptive contrastive awareness. As shown in Fig. 2 and Eq. (18), the exponential function allows easy samples (e.g., high-similarity positive pairs and low-similarity negative pairs) to retain relatively high weights, while simultaneously adjusts the contribution of hard samples through controlled weighting. This property ensures that the model focuses on both reliable and informative pairs, thereby facilitating more effective contrastive optimization and enhancing robustness.
>
> Overall, the exponential weighting strategy not only supports fine-grained differentiation across the four contrastive categories, but also enhances the ability to perform confidence-aware learning in an adaptive and theoretically grounded manner.
>
> **Q1**: *How about the adaptability of the work to heterogeneous graphs? I recommend to discuss this issue in brief.*
>
> **Reply**: Thank you for your insightful comment. Although our method does not explicitly incorporate heterogeneous node or edge types, its architectural design exhibits strong potential for application to heterogeneous graphs. Specifically, the Hybrid-Collaborative Augmentation (HCA) module jointly leverages node-level and edge-level embeddings through a flexible similarity fusion mechanism, which does not assume uniform node semantics and can naturally adapt to scenarios with diverse node types and relation patterns. Moreover, the CSADA module performs adaptive weighting based on pseudo-label confidence and sample similarity, without relying on type-specific assumptions. This design enables our model to capture fine-grained structural and semantic information, making it inherently compatible with heterogeneous graph characteristics such as multi-type interactions and complex topology.
>
> Here, we have evaluated the proposed method on datasets such as BAT and EAT, involving heterophilic structural characteristics. From a semantic perspective, heterophily graphs can be viewed as a special type of heterogeneous graphs, where the heterogeneity stems from label interactions rather than node or edge types. Compared with the representative HSAN method, our proposed RAGC achieves average improvements on the BAT and EAT datasets of $2.20$% in ACC, $2.04$% in NMI, $1.93$% in ARI, and $1.97$% in F1. The excellent performance achieved by our model on them provides supporting evidence for its potential to generalize to broader heterogeneous graph scenarios. We thank you again for this valuable suggestions. In the future, we would try to extend the proposed RAGC to general heterogeneous graph clustering.
>
> **Q2**: *The augmentation strategies for nodes and edges are from different feature sources, which might result in some problems of conflicting objectives. Please have a discussion on this issue.*
>
> **Reply**: Thank you for your valuable suggestions. In this study, indeed, the node and edge augmentation strategies are from different feature spaces, may result in potential objective conflict. To address this, we designed the Hybrid-Collaborative Augmentation (HCA) module based on multi-source information. Specifically, we construct a contrastive similarity matrix that fuses structural and attribute information, and introduce a learnable balance coefficient $\alpha$ to dynamically weigh node-level and edge-level augmentations (see Eq. (10)). Moreover, while using structural information to filter out high-frequency noise in attributes, we further guide the edge-level representation updating through both attribute and structural cues. This allows node representations to reversely influence the edge augmentation process, enabling collaborative consistency between the them. Ablation studies (see Section 3.3) in the experimental section also demonstrate that removing the HCA module significantly degrades performance, which validates the effectiveness of the hybrid augmentation strategy with collaborative interaction.
>
> Therefore, the augmentation strategies from different sources are effectively aligned through the carefully designed collaborative interaction mechanism in RAGC, achieving joint augmentation of attribute and structure information. This design improves objective consistency and enhances the discriminability and robustness of RAGC in graph embedding representation learning.

---

> ### Comment · Reviewer_zM6C · 2025-08-09
>
> This paper proposes a novel contrastive attributed graph clustering, RAGC, which mainly focuses on the issues in effective data augmentation and contrastive objective setting. This work exhibits relatively good innovation, particularly in the proposed contrastive sample adaptive-differential awareness. However, there are still some issues to be considerd, i.e., large-scale scalability should be further considered, which is important for graph clustering. Therefore, I decide to adjust the score accordingly.

---

> > ### Author Response · Authors · 2025-08-09
> >
> > Thank you. Indeed, we acknowledge that the current experiment only be conducted on medium-scale datasets, primarily due to limitations in our experimental hardware resources. However, the scalability has been fully considered in the model design. Specifically, we employ the low-frequency filtering combined with lightweight MLP-based augmentation, which avoids complex and computationally intensive operations, thereby effectively reducing computational complexity. Additionally, the K-means based pseudo-label generation process in our method is highly parallelizable, facilitating adaptation to large-scale data processing.

---

### Note · Authors · 2025-08-13

We thank chairs for processing our manscript and reviewers for their insights, with most recognizing the innovation and value of this study and giving the positive scores. Next, we would further summarize the contributions of this study and the concerns addressed in the rebuttal.

**Contributions:** The data augmentation and contrastive objective setting are two crucial factors for contrastive graph clustering. Specifically, our proposed method addresses both aspects and proposes novel strategies for them. Beyond on the single node node-level embedding augmentation, the designed hybrid-collaborative augmentation strategy achieves the edge-level embedding augmentations and models the interactions between them across various granularity. Further, we design a novel contrastive sample adaptive-differential awareness (CSADA) mechanism,  which adaptively identifies all contrastive sample pairs and differentially treats them by an innovative weight modulation function. The effectiveness of proposed method and scalability of crucial module are verified.

**Summary of Main Concerns Addressed in the Rebuttal:**
1. We explained the design principles of the key modules, including the hybrid-collaborative augmentation between node-level and edge-level,  dynamic confidence selection mechanism, and adaptive-differential awareness function in contrastive setting.
2. We analyzed the scalability of proposed method on large-scale and heterogeneous graph and provided some experimental results.
3. We justified key parameter choices, explaining the range of coefficient factors $\beta$ and $\gamma$ in adaptive-differential awareness function.

After rebuttal, we believe that the reviewers have further understood and recognized our paper.

---

### Decision · Program_Chairs · 2025-09-17

**Decision:**

Accept (poster)

**Comment:**

This paper proposes a novel robust attributed graph clustering (RAGC), which designs hybrid-collaborative augmentation from both node and edge levels and achieves the interactions between them. More importantly, the sample adaptive-differential awareness mechanism is innvatively proposed to differentially treat contrastive sample pairs  by an innovative weight modulation function, which is scalable in general contrastive learning.

The response from authors well addresses the concerns from reviewers, and after the rebuttal, four positive scores (three accept and one borderline accept) are received. All reviewers recognized its innovative idea, interesting theoretical method, and persuasive experimental results and thus reached an agreement to accept it.  Hence, I recommend it for acceptance.